# Global maps of aerosol single scattering albedo using combined CERES-MODIS retrieval

Archana Devi[1], Sreedharan K Satheesh[1,2,3]

[1] Centre for Atmospheric and Oceanic Sciences, Indian Institute of Science, Bengaluru, India
[2] Divecha Centre for Climate Change, Indian Institute of Science, Bengaluru, India
[3] DST Centre of Excellence in Climate Change, Indian Institute of Science, Bengaluru, India

*Correspondence to:* Archana Devi (archana.shiva13@gmail.com)

**Abstract.** Single Scattering Albedo (SSA) is a leading contributor to the uncertainty in aerosol radiative impact assessments. Therefore accurate information on aerosol absorption is required on a global scale. In this study, we have applied a multi-satellite algorithm to retrieve SSA (550 nm) using the concept of 'critical optical depth.' Global maps of SSA were generated following this approach using spatially and temporally collocated data from Clouds and the Earth's Radiant Energy System (CERES) and Moderate Resolution Imaging Spectroradiometer (MODIS) sensors on board Terra and Aqua satellites. Limited comparisons against airborne observations over India and surrounding oceans were generally in agreement within ±0.03. Global mean SSA estimated over land and ocean is 0.93 and 0.97, respectively. Seasonal and spatial distribution of SSA over various regions are also presented. Sensitivity analysis to various parameters indicate a mean uncertainty around ±0.044 and shows maximum sensitivity to changes in surface albedo. The global maps of SSA, thus derived with improved accuracy, provide important input to climate models for assessing the climatic impact of aerosols on regional and global scales.

## 1 Introduction

Atmospheric aerosols play a significant role in the Earth's radiation budget (IPCC, 2013). The climatic impact of aerosols depends on their absorption and scattering properties, quantified by Single Scattering Albedo (SSA). Even a slight reduction in SSA can change the aerosol radiative forcing from cooling to warming, depending on the underlying surface albedo (Kaufman et al., 2001; Chand et al., 2009). However, the lack of an accurate global aerosol absorption database has led to SSA being the largest contributor to the total uncertainty in aerosol radiative impact assessment (IPCC, 2013).

The high spatio-temporal variability in aerosol properties entails the need for observations on a global scale (Dubovik et al., 2002; Levy et al., 2007; Remer et al., 2008; Hammer et al., 2018). Satellite data, despite its inherent limitation associated with an inverse problem, can provide the global perspective required in analysing spatio-temporal aerosol characteristics (Torres et al., 2002; Lenoble et al., 2013). However, it is difficult to

quantify the absorption over bright surfaces (Kaufman and Joseph, 1982; Ahn et al., 2014; Jethva et al., 2018). Hence, quantifying the aerosol absorption over land regions using satellite-based remote sensing remains a challenge even now (Torres et al., 2013; Jethva and Torres, 2019).

Fraser and Kaufman., 1985 developed a critical surface reflectance method to retrieve SSA using satellite data. Their method is based on radiative transfer simulations, which showed a particular surface reflectance for which

the top of atmosphere reflectance is independent of AOD. Upward reflectance between a clear and a hazy day over a varying surface reflectance region are used, along with radiative transfer simulations, to derive SSA. This method has been widely applied to data from various satellites to derive SSA over particular regions (Kaufman, 1987; Kaufman et al., 1990, 2001; Zhu et al., 2011; Wells et al., 2012). Seidel and Popp., 2012 have done extensive studies on the method's sensitivity to various parameters.

Various studies have ascertained the inadequacy of single-sensor data in the accurate retrieval of aerosol absorption (Kaufman et al., 2001; Zhu et al., 2011). Dawn of the A-Train satellite constellation (Anderson et al., 2005) with spatially and temporally near-collocated observations facilitates multi-satellite retrieval of aerosol absorption (Eswaran et al., 2019; Hsu et al., 2000; Hu et al., 2007, 2009; Jeong and Hsu, 2008; Narasimhan and Satheesh, 2013; Satheesh et al., 2009)  However, all these multi-sensor retrievals are in the Ultra Violet (UV)

wavelengths, and SSA is extrapolated to visible wavelengths using spectral dependence of assumed particle size distribution. Satheesh and Srinivasan (2005) defined the concept of "critical optical depth" ($\tau_c$) and introduced a method to retrieve SSA in the visible region by combining ground-based and satellite measurements. The method was validated/demonstrated over many locations, including the desert location of Solar Village in Saudi Arabia, using Aerosol Robotic Network (AERONET) data.

In this paper, we have utilized the concept of $\tau_c$ and further extended the methodology to develop the combined CERES-MODIS retrieval algorithm to derive regional and global maps of aerosol absorption (550 nm) using multi-satellite data. The "critical optical depth" method developed in this research paper shares a similar concept to the critical surface reflectance method (Fraser and Kaufman., 1985). For a particular parameter (such as surface reflectance or optical depth), there exists a critical value at which the top of atmosphere albedo/reflectance can be

considered independent of variations in that parameter. Both the methods retrieve SSA by parameterizing the critical value as a function of SSA using radiative transfer simulations. The critical reflectance method requires two-days data and large variations in surface reflectance over the region. It's suitable for retrieving daily SSA for a particular region. Whereas the critical optical method developed in this paper is suitable for retrieving monthly or seasonal global maps of SSA.

The concept of $\tau_c$, which forms the scientific basis for the development of this retrieval algorithm is illustrated in Section 2. The various steps involved in the retrieval algorithm are detailed in the Section 3, data and methodology. Section 4 presents the results and comparison with other satellite datasets. Uncertainity analysis is studied in Section 5. Comparison with aircraft measurements from various field campaigns are shown in Section 6. Comparison with AERONET data from 15 sites are shown in section 7. Summary and conclusions are provided in Section 8.

**2 Critical optical depth**

Let $\Delta\alpha$ be the difference between the top of the atmosphere (TOA) albedo and surface albedo. Then, for a particular location, with a given surface albedo, $\Delta\alpha$ variations are only due to changes in TOA albedo. The presence of absorbing aerosols over a bright surface decreases the TOA albedo. In contrast, scattering aerosols over a dark surface increase the TOA albedo. Thus, the increase (decrease) in aerosol loading due to scattering (absorbing) type of aerosols leads to an increase (decrease) in $\Delta\alpha$. The rate of change in $\Delta\alpha$ with aerosol loading is dependent on SSA.

Satheesh and Srinivasan (2005) utilized this concept to retrieve SSA in the case of absorbing aerosols over a bright surface. In a pristine atmosphere (Aerosol Optical Depth = 0) over a bright surface, the $\Delta\alpha$ is positive for solar zenith angle (SZA) = 0. Here, when absorbing aerosols become dominant, $\Delta\alpha$ decreases with an increase in aerosol optical depth (AOD) and eventually turns negative. The AOD at which $\Delta\alpha$ equals zero is defined as $\tau_c$. For a given surface albedo, $\tau_c$ is the AOD at which the scattering and absorbing effects of the aerosol cancel each other. The rate of decrease in $\Delta\alpha$ with the increase in AOD is higher when SSA is high and consequently lowers the resulting values of $\tau_c$. A radiative transfer (RT) model was then used to calculate the SSA that reproduces the same $\tau_c$, given atmospheric conditions.

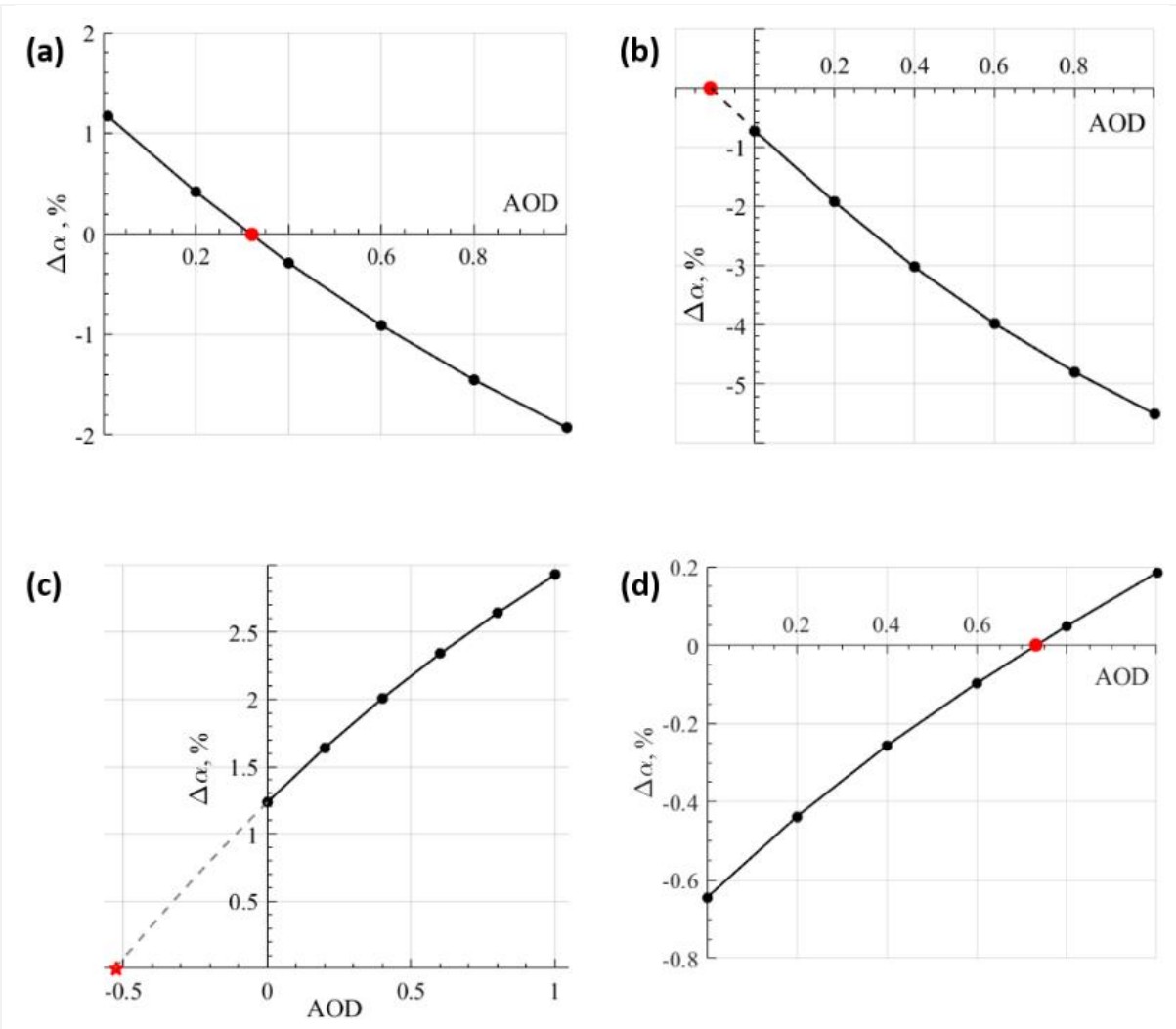

**Figure 1.** RT simulations (black dots) shows deriving $\tau_c$ (red dot) for different cases of aerosols and surfaces. For pristine conditions (AOD = 0), diurnally-averaged $\Delta\alpha$ is negative for bright surfaces and positive for dark surfaces. An increase in aerosol loading by absorbing (scattering) type of aerosol leads to decrease (increase) in TOA albedo. **(a)** Absorbing aerosols above a dark surface; **(b)** Absorbing aerosols above a bright surface; **(c)** Scattering aerosols above a dark surface; **(d)** Scattering aerosols above a bright surface.

In this paper, the concept of $\tau_c$ is extended to retrieve SSA for all scenarios of surfaces (dark and bright) and aerosols (absorbing and scattering). For AOD less than 1, $\Delta\alpha$ is almost linearly dependent on AOD. Then $\tau_c$ is mathematically the x-intercept when parameterizing the linear relationship.

5    Figure 1 shows the estimation of $\tau_c$ for four different scenarios. Details of these RT simulations are given in Section 3.2. Unlike Satheesh and Srinivasan (2005), where simulations were carried out for SZA = 0, here the $\Delta\alpha$ is diurnally averaged. Therefore, it is possible to have negative $\Delta\alpha$ for AOD = 0 over relatively bright surfaces. It is difficult to retrieve SSA where the slope of regression line is close to zero.

## 3 Data and methodology

The Combined CERES-MODIS retrieval algorithm consists mainly of two steps: (1) determining $\tau_c$ using MODIS and CERES data for a location, and (2) estimation of SSA that reproduces the same $\tau_c$ for the associated atmospheric conditions and surface albedo of that particular location. Figure 2 shows the flowchart illustrating the combined CERES-MODIS retrieval algorithm.

TOA and surface fluxes, used to determine $\Delta\alpha$, are obtained from CERES SYN1deg-day (Edition 4.1) (Wielicki et al., 1996; Rutan et al., 2015). To avoid angular dependence of fluxes, the diurnally averaged flux data product from CERES is used, which is available only at 1° resolution. Hence, other satellite data sets in this study are also used at the same spatial resolution. AOD and total columnar water vapor are obtained from the MODIS Daily Global Product (MxD08_D3 version 6.1). MODIS retrieves columnar AOD at 550 nm using two different types of algorithms – "Dark Target" (Levy et al., 2007, 2013) and "Deep Blue" (Hsu et al., 2004, 2006; Sayer et al., 2013). Dark target retrieves AOD over both land and ocean, whereas deep blue retrieves only over land. In this study, we have used a combined dark target and deep blue product.

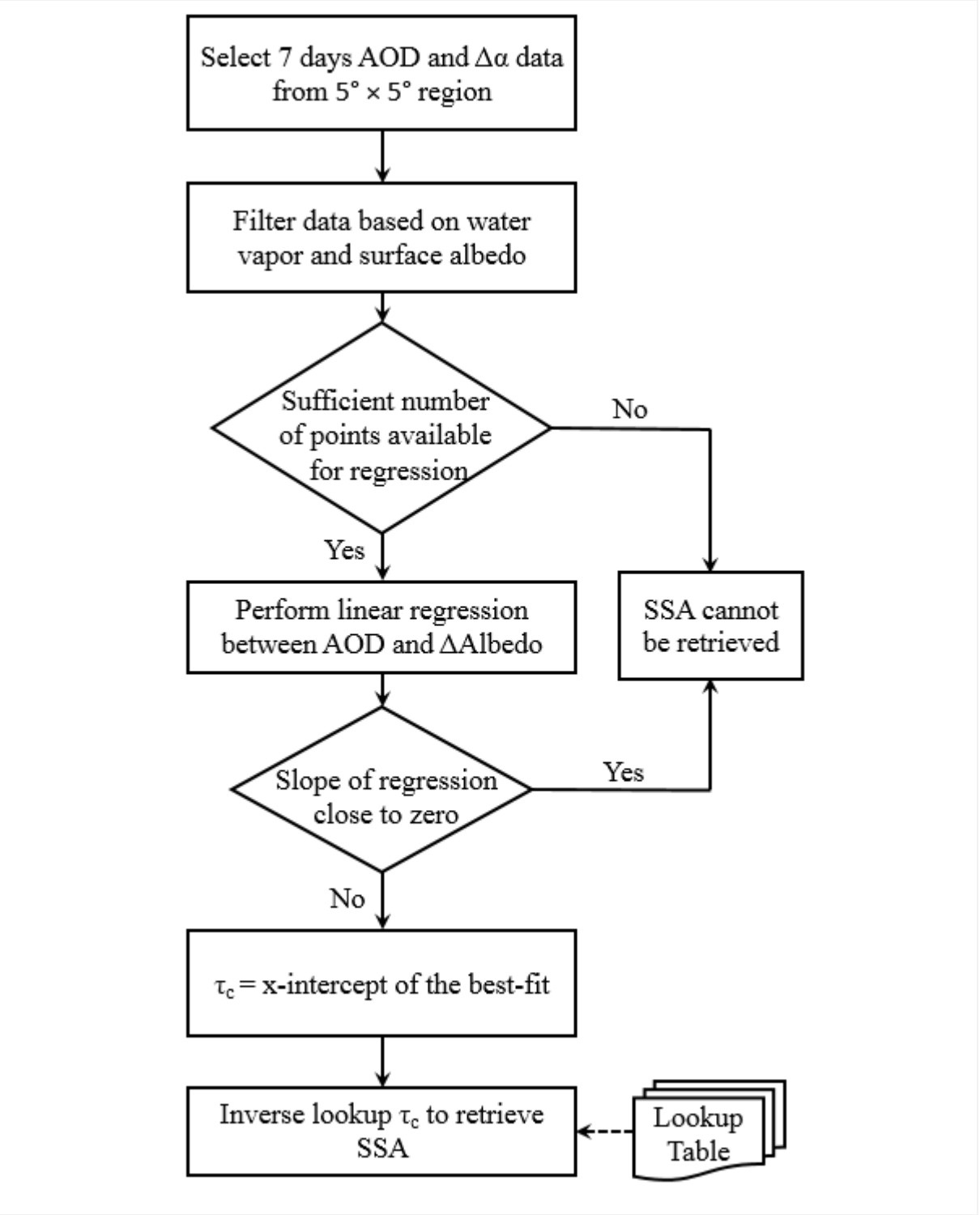

**Figure 2.** Flowchart depicting the steps involved in combined CERES-MODIS retrieval of SSA for a particular location.

## 3.1 Determining the critical optical depth

The first step for retrieval is to determine $\tau_c$ by linear regression analysis between $\Delta\alpha$ vs. AOD as shown in Fig. 3. The x-intercept of the resultant line of best fit (i.e., the AOD at which $\Delta\alpha = 0$) provides the value of $\tau_c$. CERES

and MODIS daily data are at 1° resolution, and SSA is retrieved for each 1° × 1° grid. In order to have adequate number of points for a meaningful regression analysis, it was required to use data over a larger interval (temporal and spatial) - whose extent is large enough to get a statistically significant fit but small enough to ensure insignificant variations in SSA. Thus, to determine $\tau_c$ for a given pixel, seven days of data from its surrounding

5° × 5° region has been considered. This data is further constrained based on surface albedo and water vapor. Only those pixels in this region having surface albedo within ± 0.025 and water vapor within ± 0.25 cm of the given pixel are considered for regression analysis. These constraints ensure that the $\tau_c$ determined from the best fit is dependent only on SSA and not affected by changes in surface albedo and water vapour. Figure 3a shows an example of regression with a positive correlation coefficient over the Arabian Sea. This can happen over regions

of low surface albedo and the dominance of scattering aerosols. Figure 3b is an example of regression analysis with a negative correlation coefficient obtained over Sahara in the presence of dust aerosols.

The above procedure is repeated for all pixels, where data from the surrounding 5° × 5° region is used to determine $\tau_c$ for each pixel. For the regression analysis, points which are outside one standard deviation are considered as outliers. Line of best fits with a slope close to zero yields extreme $\tau_c$ values (very high positive/very low negative).

In such cases, we did not attempt a retrieval. A significance test on the correlation coefficient between AOD and $\Delta$Albedo is performed with a 0.05 significance level. Only those $\tau_c$ values obtained through regressions that are statistically significant at 95% confidence level are utilized further to retrieve SSA.

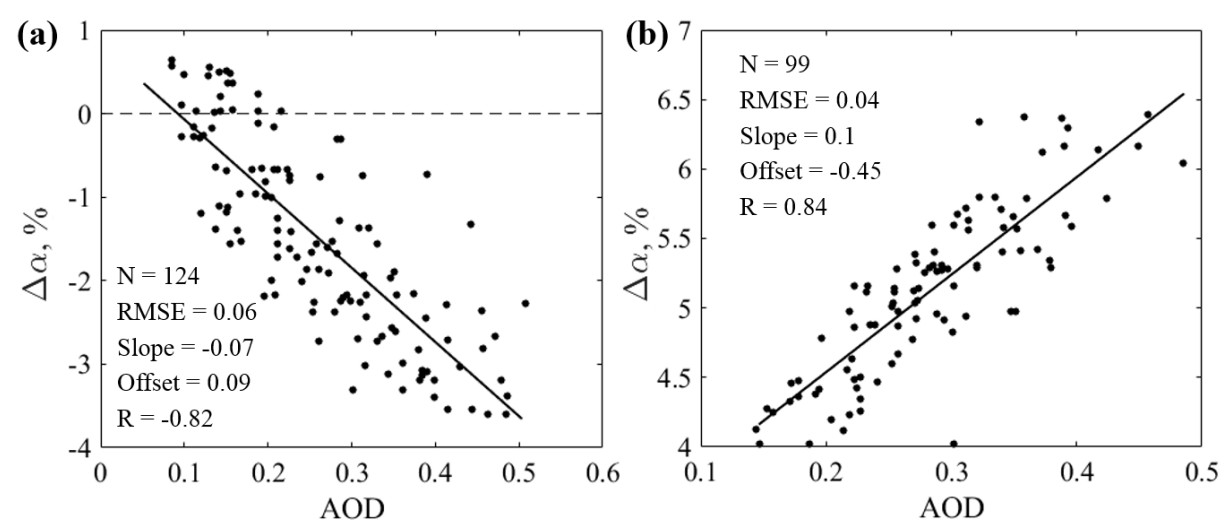

**Figure 3.** Sample scatterplots between MODIS AOD and CERES $\Delta\alpha$. The solid lines represent the best-fits for **(a)** absorbing aerosols above the Sahara and **(b)** scattering aerosols above the Arabian Sea. $\tau_c$ (AOD at which $\Delta\alpha$ is zero) is the x-intercept of the best-fit line.

The final product of this step is a 360 × 180 matrix that stores $\tau_c$ value corresponding to each 1° pixel. In these matrices, not all points would have a $\tau_c$ value owing to the insufficient number of points available for regression, either due to cloud-masking or large variations in surface albedo over the land. At least seven days of data is required to perform a statistically significant fit to compute $\tau_c$ and retrieve SSA The next step in the procedure is to estimate SSA from these $\tau_c$ values using an inverse lookup table (LUT) approach.

**3.2 Retrieval of SSA**

Since the objective of this study is to retrieve SSA globally, look-up-tables (LUTs) were developed to reduce the computation time and avoid repeated RT simulations. The aerosol models available in OPAC (Optical Properties of Aerosols and Clouds), developed by Hess et al., (1998), are given as input to SBDART (Santa Barbara DISORT Atmospheric Radiative Transfer) model (Ricchiazzi et al., 1998) to simulate TOA fluxes. Specifications of the models used are shown in Table S5, S6, S7 and S8.

The RT computations were carried out to obtain the diurnally averaged (SZA: 0° to 84°) TOA and surface fluxes using 16 radiation streams and spectrally integrated over the shortwave region (0.3 to 5 μm). For a particular case of surface albedo, water vapor, and SSA, AOD is varied from 0 to 1 in steps of 0.2 to generate its corresponding diurnally averaged $\Delta\alpha$. Then a linear fit is performed between AOD and simulated $\Delta\alpha$ to determine $\tau_c$. For each aerosol model a three-dimensional LUT that stores $\tau_c$ for different combinations of surface albedo, water vapor, and SSA have been developed. The LUT is indexed by 11 values of surface albedo (0 to 0.5, increments of 0.05), 17 values of water vapour (0 to 8 cm, increments of 0.5 cm) and 10 values of SSA (0.8, 0.83, 0.85, 0.87, 0.9, 0.92, 0.95, 0.97, 0.99, and 1). A total of 89760 RT simulations were performed in the present study.

The next step is to estimate SSA from $\tau_c$ using the LUT. For a given surface albedo and water vapor of that pixel, we find the SSA associated with its determined $\tau_c$. An inverse lookup operation is performed on LUT by linear interpolation between the nearest two indices. The aerosol model (LUT) selected for retrieval is based on geographic location (ocean/land, surface albedo) and aerosol loading. Details of aerosol model selection is shown in Fig S4 and S5. SSA is estimated for each available $\tau_c$ values of a pixel and then averaged to compute the seasonal mean SSA.

**4 Results and discussion**

Fig. 4 shows the seasonal-mean global maps of SSA (550 nm) retrieved by the combined CERES-MODIS algorithm for the five years of 2014-2018. Data are averaged for different seasons: DJF (December-January-February), MAM (March-April-May), JJA (June-July-August), and SON (September-October-November).

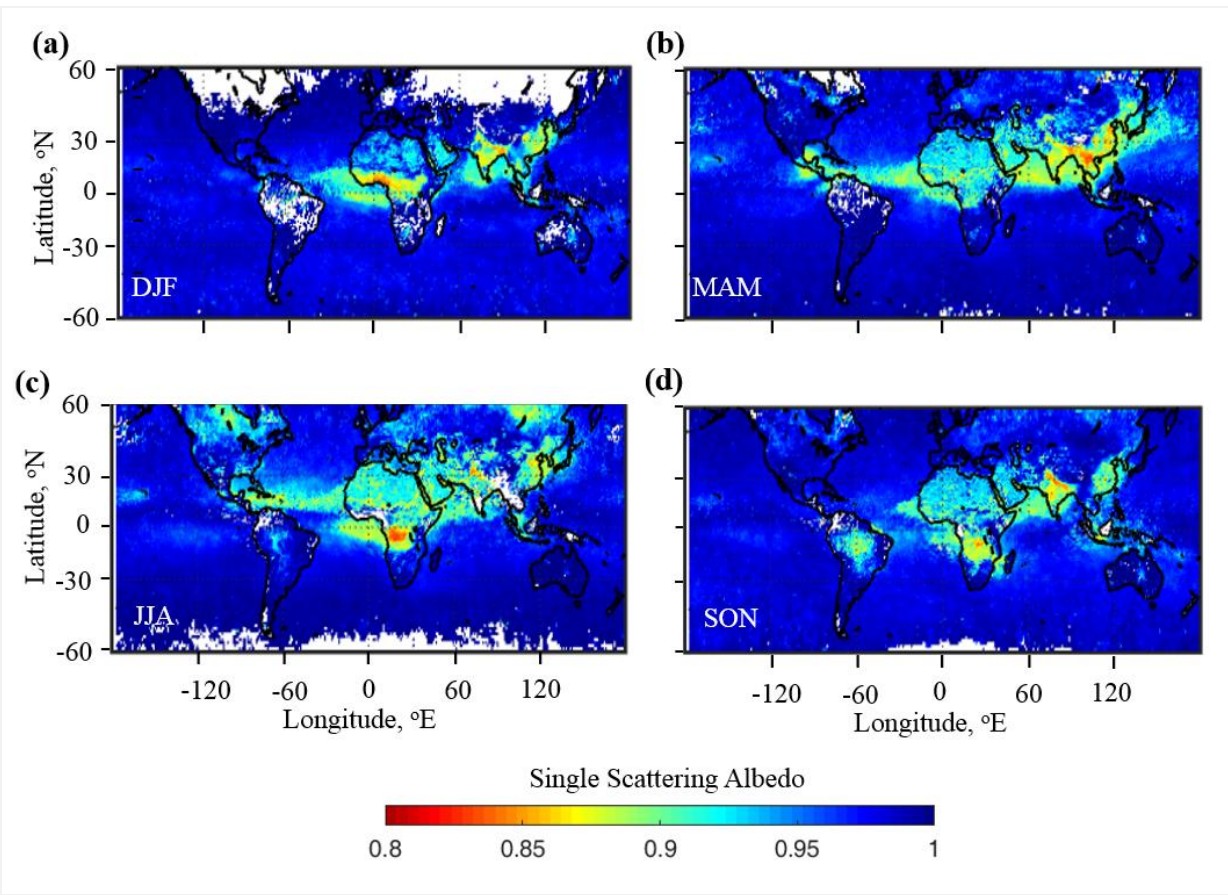

**Figure 4.** Seasonal mean SSA maps for the period of 2014-18 retrieved by the combined CERES-MODIS.

The retrieved SSA dataset (550 nm) was compared with other widely used global SSA datasets – OMI SSA (500 nm) and climatological POLDER SSA (565 nm). OMAERUVd V3 (Torres et al., 2007; Torres et al., 2013; Ahn et al., 2014) for the corresponding period are shown in panels a, c, e, and g in Fig 5. And POLDER v1.2 Level 3 (Dubovik et al., 2011, 2014, 2021) climatological seasonal mean SSA maps are shown in panels b, d, f, and h in Fig 5. For a generalized qualitative comparison, we can assume that SSA does not vary much for the small 50 nm spectral difference between CERES-MODIS and OMI SSA. (Zhu et al., 2011; Jethva et al., 2014).

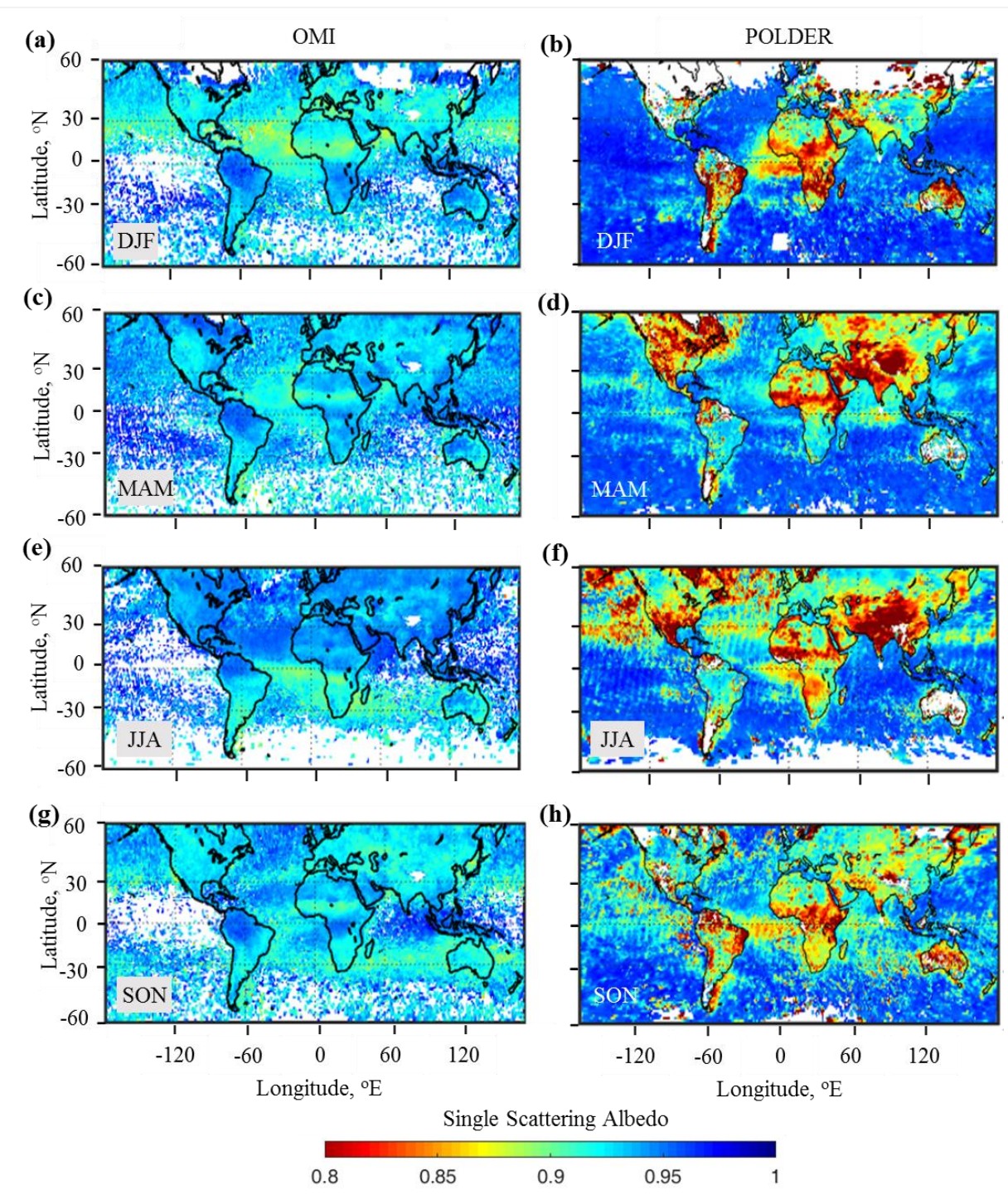

**Figure 5.** Seasonal mean SSA maps of OMI (500 nm) and POLDER (565 nm) in panels a,c,e,g and b,d,f,h respectively.

From a quick comparison between Fig 4 and Fig S2 SSA maps, the following points can be noted:

- Over the ocean, OMI retrieves SSA only for regions with high values of UVAI, leading to large data gaps. In comparison, we can notice that CERES-MODIS and POLDER have better data coverage on a global scale. In the CERES-MODIS maps, the absence of data is mostly due to the unavailability of MODIS AOD.

- The Global Ocean, a relatively dark surface covering more than 70% of the Earth's surface, plays a significant role in determining global aerosol radiative forcing effects. Therefore, the better data coverage over oceans by the CERES-MODIS and POLDER provides better input for radiative forcing calculations.

- CERES-MODIS maps capture a wider range of SSA values. Regions with very low SSA can easily be identified as the sources of absorbing aerosols. OMI SSA values are mostly above 0.9 and do not clearly capture the sources and transport of absorbing aerosols.

- OMI SSA values are more accurate in the UV wavelengths since SSA is primarily retrieved in the UV regions and extrapolated to visible wavelengths using aerosol models. Whereas CERES-MODIS retrieves SSA directly at 550 nm, hence is more accurate for SSA values in the visible wavelengths.

- Large variations in SSA can be observed between CERES-MODIS and POLDER, especially over land where the aerosol loading is less. POLDER SSA retrievals are more accurate for higher aerosol loading. Chen et al. 2020 has shown that POLDER SSA (670 nm) comparison with AERONET significantly improves with correlation coefficient increasing from 0.321 to 0.814 and RMSE decreasing from 0.056 to 0.029 for AOD greater than 1.5.

- Over the land, POLDER shows very low SSA values (< 0.85), thus indicating the presence of highly absorbing aerosols even over less polluted regions. OMI values are around 0.9 over land and do not clearly identify the presence of absorbing aerosols. Whereas SSA values are within reasonable range over land as retrieved by the CERES-MODIS method – high SSA values over relatively pristine regions, lower SSA values over sources and transport of absorbing aerosols.

- Seasonal trends in forest fire can be noticed in POLDER maps and distinctly identifiable in CERES-MODIS SSA maps. Every year forest fires are common in specific seasons in Canadian and Russian Boreal forests (JJA), Amazon forest (SON) and South African forest (JJA and SON).

-   The Indo-Gangetic plain (IGP) is a densely populated region spotted with several coal-based thermal power plants and seasonal stubble burning. Low SSA values are retrieved by both POLDER and CERES-MODIS over IGP. Whereas OMI shows values around 0.9 throughout the year. Similar pattern can be observed over Eastern China, one of the most highly polluted industrial region.

5    From the above points, we can draw conclusions about the advantages of each dataset. OMI, CERES, and MODIS instruments are still operational, whereas POLDER datasets are available only till 2013. OMI datasets are more suitable for UV wavelengths, whereas the CERES-MODIS SSA dataset provides more accurate SSA over visible wavelengths. OMI provides operational daily global SSA maps, whereas the CERES-MODIS algorithm is more suitable for obtaining monthly/seasonal global SSA maps. Over the ocean, the POLDER dataset has more coverage than OMI and identifies the transport of aerosols across the oceans. Hence, POLDER SSA and CERES-MODIS SSA can be used for studying SSA values over the ocean in the UV and visible wavelengths, respectively. Over the land, OMI retrieves high SSA values, whereas POLDER shows very low SSA values even over relatively pristine regions. Hence, the CERES-MODIS dataset retrieves reasonable SSA values over both polluted and less polluted regions for visible wavelengths.

15    Global mean SSA retrieved by combined CERES-MODIS over land and ocean is 0.93 and 0.97, respectively (OMI: 0.94 and 0.94; POLDER: 0.88 and 0.94). Accurate SSA estimations are also required over regions of interest such as deserts, oceans, biomass-burning forests, and highly polluted industrial areas. Hence, seasonal mean SSA values retrieved by the combined CERES-MODIS algorithm, OMI, and POLDER are reported, in table S2, for major regions of interest as shown in Fig S1 and Table S1.

20    **5 Uncertainty Analysis**

Table 1 identifies the major sources of error in the retrieval and summarizes their individual contribution. Uncertainty in the retrieved SSA was estimated by calculating retrieval sensitivities to perturbations in the possible error sources. The range of perturbation was based on published literature or reasonable assumptions for possible variations. Also, since SSA is computed from tauC which depends on the slope of the regression, uncertainties 25    due to each error source was computed by perturbing them for different cases of SSA (0.8 to 1 in steps of 0.01). For example, uncertainties in surface albedo were calculated by perturbing it by ±0.01 for different cases of surface albedo (dark to bright: 0.05 to 0.5 in steps of 0.05) and SSA (absorbing to scattering: 0.8 to 1 in steps of 0.01). The mean value of the uncertainties obtained from all these cases is shown as retrieval uncertainty in Table 1.

**Table 1**. Estimates of the uncertainty in retrieved SSA

| Parameter | Input Uncertainty | Retrieval Uncertainty |
|---|---|---|
| Surface albedo | ±0.01 | ±0.03 |
| AOD | 20% ±0.05 (land) 5% ±0.03 (ocean) | ±0.02 |
| Ångström exponent | ±0.4 | ±0.01 |
| Refractive index | ±0.01 | ±0.01 |
| Aerosol height | ±1 km | ±0.01 |
| Aerosol type | Smoke vs dust | ±0.01 |
| Residual of fit | ±0.05 | ±0.02 |

Uncertainty in shortwave integrated surface albedo from CERES results in the maximum uncertainty in SSA of ±0.03. MODIS retrieved aerosol optical depth contains considerable uncertainties due to assumed aerosol models (Jeong et al., 2005). The MODIS aerosol optical depth uncertainty is 20% ±0.05 over land (Chu et al., 2002) and 5% ±0.03 over the ocean (Remer et al., 2002). The corresponding error in our retrieval is ±0.02. For a typical variation of Ångström exponent (±0.4) and imaginary part of the refractive index (±0.01), the uncertainties vary depending on the surface albedo and are mostly around ±0.01.

Changes in aerosol height can vary the TOA radiances due to Rayleigh scattering interactions, which depend on pressure. Sensitivity to aerosol height was estimated by conducting a synthetic retrieval of SSA over a range of aerosol height values and perturbations from those heights. The average uncertainty observed for an aerosol height variation of ±1 km was ±0.01. Many methods have been developed for detecting aerosol type, especially smoke vs. dust, to improve the uncertainties of various AOD and SSA retrievals.

Uncertainties due to possible variations on scales of the regions used for linear fitting were estimated as residuals of the fit. The uncertainty on the linear intercept is spatially dependent and is mostly around ±0.02, with higher values for those combinations having a slope close to zero during the regression. For highly correlated cases (i.e., correlation coefficient $|r| > 0.5$), the probability of obtaining a slope close to zero is ~20% over the ocean and <5% over land. These cases are mostly formed over regions where AOD variations are less. Regions having large variations in AOD values have lower uncertainty due to residual fit. Adding in quadrature, the total uncertainty estimated for the CERES-MODIS algorithm is around ±0.044.

Overall, the algorithm is most sensitive to variations in surface albedo, followed by higher sensitivity towards AOD values used in the linear fit. Seaonal mean maps of surface albedo are shown in Fig S3. The uncertainties are higher for scattering aerosols over bright surfaces and absorbing aerosols above dark surfaces. Sensitivity to water vapor is almost negligible, except in very few cases where the uncertainty is $\pm$ 0.008. The CERES-MODIS algorithm is most effective over regions with large AOD variations and less surface albedo variations.

**6 Comparison with airborne observations**

For the comparison of columnar SSA values thus retrieved, we have used aircraft-based measurements of SSA from three campaigns: South West Asian Aerosol Monsoon Interactions (SWAAMI), Regional Aerosol Warming Experiment (RAWEX), and SWAAMI-RAWEX, to obtain column-integrated SSA. Available data points over India and adjoining oceanic regions (Arabian Sea and Bay of Bengal) from these field campaigns were compared with the retrieved SSA.

Babu et al. (2016), as part of RAWEX (Moorthy et al., 2016), derived SSA at 520 nm from aircraft measurements of scattering and absorption coefficients over the Indo-Gangetic Plain (IGP) and Central India during winter 2012 and spring/pre-monsoon 2013. Various measurements of aerosol properties were carried out in an instrumented Beechcraft B200 aircraft of the National Remote Sensing Centre, India. Manoj et al. (2019) estimated vertical profiles of SSA during the SWAAMI campaign conducted during monsoon (June - July) 2016 over IGP, Arabian Sea, and Bay of Bengal. Aerosol scattering coefficients were measured aboard the Facility for Airborne Atmospheric Measurements (FAAM) BAe-146 aircraft. Vaishya et al. (2018) estimated vertical profiles of SSA (520 nm) using an instrumented aircraft, Beechcraft B200, during SWAAMI-RAWEX campaign (June 2016). Instrument design and calibration were based on Anderson et al., 1996 and its application for Indian field experiments was as described by Nair et al., 2009. Uncertainties in the scattering coefficient measurement by nephelometer are ~±10%, as reported by Anderson et al., 1996. As stated by Babu et al., 2016 uncertainties in the columnar SSA values estimated from RAWEX aircraft measurements depend mainly on instrumental uncertainties, sampling errors, and large spatial averaging.

Retrieved SSA, for the same period as the campaign, over a 2°×2° region around the campaign location was utilized for comparison. Figure 4 shows the comparison of collocated aircraft measurements and CERES-MODIS retrieved SSA. The ideal 1:1 case (solid line), the absolute difference of 0.03 (dotted lines), and regression coefficients are also provided.

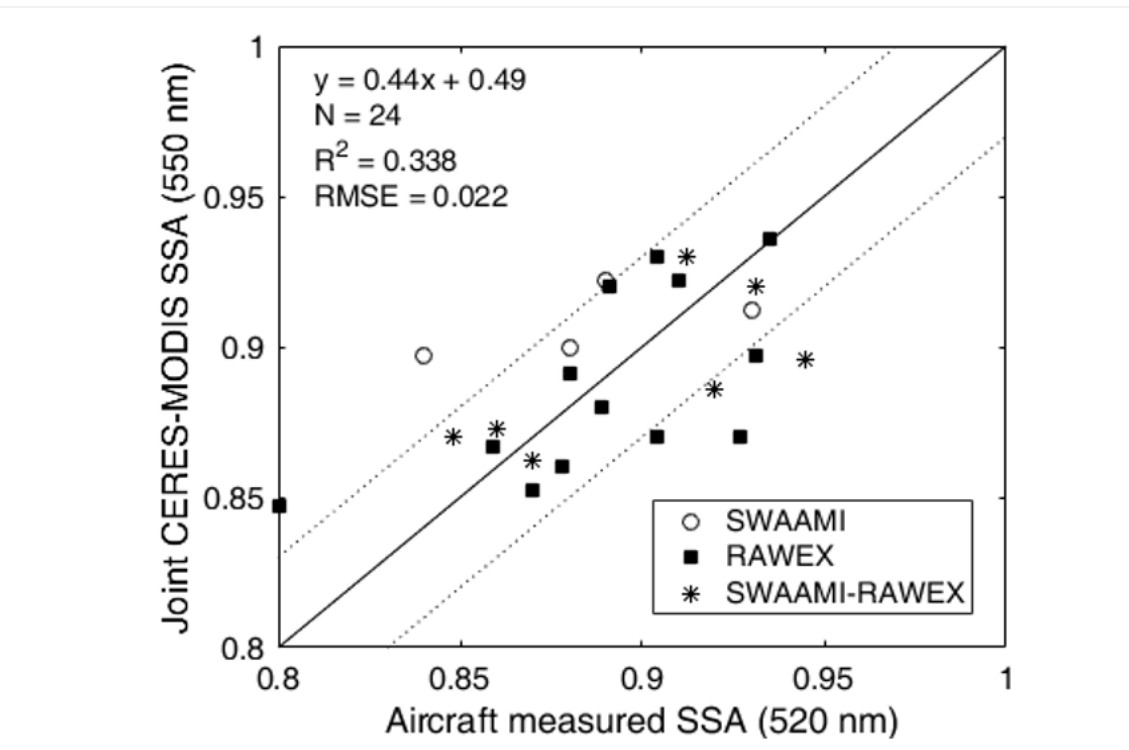

**Figure 6.** Comparison of combined CERES-MODIS SSA with aircraft measurements during SWAAMI, RAWEX, and SWAAMI-RAWEX campaigns. The solid line shows the ideal 1:1 case and dotted lines represent the absolute difference of 0.03.

Most of the points were within the absolute difference of 0.03. However, there are few exceptions. SSA values over the Bay of Bengal during SWAAMI campaign were reported as $0.84 \pm 0.07$ during June-July by Manoj et al. (2019), whereas CERES-MODIS retrieves a higher SSA of ~0.89 for the same time period. This large variation could be due to frequent cloud cover during the monsoon season, leading to fewer SSA points retrieved over the ocean and land. SSA estimated over Nagpur in Central India during RAWEX is ~0.8, while CERES-MODIS retrieves ~0.85. This inconsistency is due to the large surface albedo variations (standard deviation >0.05) over Central India, which leads to fewer points available for retrieval. Except for few such cases, most of the other points lie within an absolute difference of 0.03.

For comparison purposes, many previous studies have used ground-level SSA data from AERONET obtained through inversion methods (Zhu et al., 2011; Jethva et al., 2014). Even in this study, only very few points were available for comparsion due to the limited number of direct measurements of columnar SSA. Despite this limitation, this comparison exercise provided confidence to generate global maps of SSA following this method.

## 7 Comparison with AERONET data

The Aerosol Robotic Network is a ground-based worldwide federated network of Cimel Sun photometers that measure extinction AOD from direct Sun measurements (Holben et al., 1998). The spectral diffuse sky radiations measured at different angles are inverted in conjunction with direct Sun measurements to derive the spectral SSA (440, 675, 870, and 1020 nm) and size distribution (Dubovik and King., 2000). The estimated uncertainty in retrieved SSA is largely attributed to the uncertainties in instrument calibration and is within 0.03 for AOD (440 nm) larger than 0.4. (Dubovik et al., 2000,2002).

AERONET version 3, level 2.0 monthly average values from selected sites were compared with corresponding CERES-MODIS SSA data. Sites were chosen to represent various types of aerosols following that of Giles et al., 2012. The location of the sites is shown in Fig S2 and Table S3. Scatter plots of comparison of AERONET SSA and CERES-MODIS SSA are shown in Fig 7. AERONET SSA at 550 nm was estimated by interpolation between the values at 440 and 675 nm.

Most AERONET SSA values are above 0.85, even in case of biomass burning aerosols. For dust type of aerosols (sites: Capo_Verde, Dakar, and Banizoumbaou) and mixed type of aerosol (sites: SEDE_BOKER, Kanpur, Xiang He and Illorin) as shown in Fig 7a and 7b respectively, the AERONET and CERES-MODIS data shows good agreement. For urban (sites: GSFC, Mexico_city, Shirahama, Ispra, and Moldova) and biomass (sites: Alta_Floresta, Lake_Argyle, and Mongu), only very few data were available during the study period of 2014-18 as shown in Fig 7 panels c and d. Data points combined from all the sites are plotted together in Fig 7e, showing a RMSE of 0.026. Overall, the resulting comparisons are agreeable within the uncertainties of both AERONET and CERES-MODIS datasets.

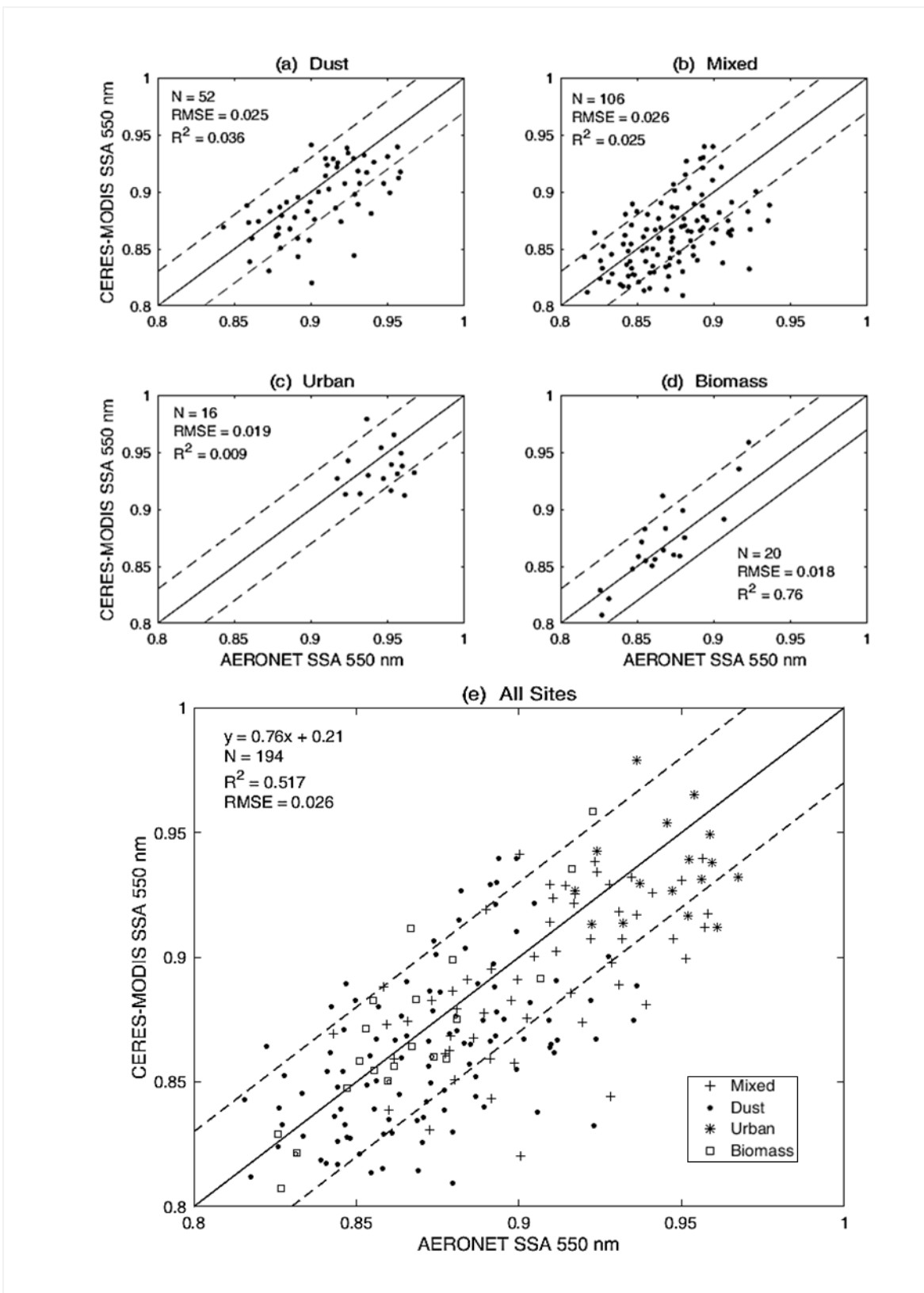

**Figure 7.** CERES-MODIS SSA (550 nm) vs. AERONET SSA (550 nm) for various AERONET sites classified based on the type of aerosols (Giles et al., 2012)

## 8. Summary and Conclusions

Global maps of aerosol absorptions were generated using the newly developed combined CERES-MODIS algorithm based on the concept of critical optical depth. The CERES-MODIS dataset was compared with OMI and POLDER SSA datasets. The retrieved SSA values were also compared with available aircraft measurements over India and surrounding oceanic regions, which showed that most retrieved SSA values are within $\pm$0.03. We showed that the combined CERES-MODIS algorithm better captures the spatial and seasonal variations in aerosol absorption and the resultant maps provide an improved global SSA database with fewer data gaps. Global mean SSA was estimated to be 0.93 and 0.97 over land and ocean, respectively. Sensitivity analysis to various parameters indicate a mean uncertainty around $\pm$0.044 and shows maximum sensitivity to changes in surface albedo. The algorithm is shown to be the most effective over regions with large AOD variations and less surface albedo variations. Comparison with SSA from 15 AERONET sites showed an acceptable agreement between AERONET and CERES-MODIS SSA within their uncertainties. These global maps provide valuable input to models for assessing the aerosol-climate impacts on both regional and global scales.

**Data Availability**

MODIS and CERES data used in this study are available at https://asdc.larc.nasa.gov/. POLDER GRASP datasets are available at https://www.grasp-open.com/products/. AERONET station data were taken from https://aeronet.gsfc.nasa.gov/. The combined CERES-MODIS datasets are available upon request from the corresponding author.

**Author Contributions**

SKS conceptualized the method. AD developed the algorithm, carried out the simulations, and analyzed the data. AD wrote the manuscript with revisions from SKS.

**Competing interests**

The authors declare they have no conflict of interest.

**Acknowledgment**

The authors gratefully acknowledge the Atmospheric Science Data Center (ASDC) at NASA's Earth Observing System Data and Information System (EOSDIS) Distributed Active Archive Centers (DAACs) for providing MODIS, OMI, and CERES data products used in this study. The PARASOL/GRASP results are generated by Laboratoire d'Optique Atmosphérique and Cloudflight Austria GmbH with the GRASP-OPEN software. We would like to thank the following principal investigators and their staff for maintaining the following sites: Phillipe Gouloub (Capo Verde), Didier Tanré (Dakar and Banizoumbou), Jean Louis Rajot (Banizoumbou), Arnon Karnieli (SEDE BOKER), Brent Holben (Kanpur, GSFC, Mexico City, Shirahama, Moldova, Alta Floresta and Mongu Inn), S N Tripathi (Kanpur), Pucai Wang and Xiangao Xia (XangHe), Rachel T Pinker (Ilorin), Itaru Sano (Shirahama), Giuseppe Zibordi (Ispra), Alexander Aculinin (Moldova), Paulo Artaxo (Alta Floresta) and Ian Lau (Lake Argyle). In addition, one of the authors (S. K. Satheesh) acknowledges the JC Bose Fellowship awarded to him by SERB-Department of Science and Technology, New Delhi. This study is supported by "Tata Education and Development Trust." We thank the Editor and the two anonymous reviewers for their valuable feedback and suggestions that have vastly improved the manuscript.

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
