# Peer review of "Global maps of aerosol single scattering albedo using combined CERES-MODIS retrieval"

_Atmospheric Chemistry and Physics, 2021_

## Author Response (AR3)

**Response to reviewer's reply**

MS no: ACP-2021-521

We thank the reviewer for the suggestions. We have revised the manuscript following the corrections suggested by the reviewer. This document outlines the reviewer's comments (in **bold-blue**), followed by the author's responses and changes made in the revised manuscript. A marked-up version of the manuscript showing the revisions is appended to this response file.
* * *
**Comment:**
**The authors have addressed my remaining concerns in this version. I recommend publication following these quite minor corrections.**

**Response:** We thank the reviewer for the feedback. We have revised the manuscript following the suggestions.

**Comment:**
**1. Page 9 line 5: I think the first "500" should read "550" (the CERES data are at 550 nm).**

**Response:** Thank you. It's corrected as 550 nm.

**Comment:**
**2. Page 9 line 7: A reference is needed for the POLDER data set used. Ideally a version number and a paper citation. The version number will depend on what the authors are using. The paper should probably be Dubovik 2014 or Chen 2020. Also, I am not sure what "POLDER 1-2" means here, this should be checked and corrected. If the authors are referring to the instrument, from the time period they are using POLDER 3 data and not POLDER 1 or 2.**

**Response:** The "POLDER 1-2" was a typing mistake. It was meant to be "POLDER v1.2". Thank you for mentioning this. It is now corrected. We have also added the relevant citations.

**Additions to the manuscript** (underlined)

Page 9 Lines 7-9: And POLDER v1.2 Level 3 (Dubovik et al., 2011, 2014, 2021) climatological seasonal mean SSA maps are shown in panels b, d, f, and h in Fig 5.

**References**

Dubovik, O., Herman, M., Holdak, A., Lapyonok, T., Tanré, D., Deuzé, J. L., Ducos, F., Sinyuk, A., and Lopatin, A.: Statistically optimized inversion algorithm for enhanced retrieval of aerosol properties from

spectral multi-angle polarimetric satellite observations, Atmos. Meas. Tech., 4, 975–1018, https://doi.org/10.5194/AMT-4-975-2011, 2011.

Dubovik, O., T. Lapyonok, P. Litvinov, et al.: GRASP: a versatile algorithm for characterizing the atmosphere, SPIE: Newsroom, Published Online: September 19, 2014. doi:10.1117/2.1201408.005558

Dubovik, O., Fuertes, D., Litvinov, P., Lopatin, A., Lapyonok, T., Doubovik, I., Xu, F., Ducos, F., Chen, C., Torres, B., Derimian, Y., Li, L., Herreras-Giralda, M., Herrera, M., Karol, Y., Matar, C., Schuster, G. L., Espinosa, R., Puthukkudy, A., Li, Z., Fischer, J., Preusker, R., Cuesta, J., Kreuter, A., Cede, A., Aspetsberger, M., Marth, D., Bindreiter, L., Hangler, A., Lanzinger, V., Holter, C., and Federspiel, C.: A Comprehensive Description of Multi-Term LSM for Applying Multiple a Priori Constraints in Problems of Atmospheric Remote Sensing: GRASP Algorithm, Concept, and Applications, Front. Remote Sens., 0, 23, https://doi.org/10.3389/FRSEN.2021.706851, 2021.

**Comment:**

**3. Page 12 line 16: I'd add the POLDER averages too here, for completeness.**

**Response:** POLDER averages have been added.

**Additions to the manuscript (**underlined**)**

Page 12 Lines 15-16: Global mean SSA retrieved by combined CERES-MODIS over land and ocean is 0.93 and 0.97, respectively (OMI: 0.94 and 0.94; POLDER: 0.88 and 0.94).

**Comment:**

**4. Page 13 line 8: One of my previous review comments was that, if the authors' uncertainty estimates for SSA are added in quadrature, you come up with about 0.044. This number has now been put in the manuscript, but doesn't appear to be in the mean paper. I think a sentence to this effect should be added somewhere around here as well, otherwise a reader may wonder where this number comes from.**

**Response:** In the last revised manuscript, we did mention it there in the uncertainty analysis section. *Page 13 Lines 19-20: Adding in quadrature, the total uncertainty estimated for the CERES-MODIS algorithm is around ±0.044.*

It was shown in the track changes file, which was uploaded separately and also appended to the response file. But we missed out to mention it in the response to reviewer.

**Comment:**

**5. Angstrom should be typeset as Ångström throughout.**

**Response:** Done. All occurences were typset as Ångström.

[revised manuscript text omitted]

---

## Author Response (AR4)

**Response to reviewer #1**

MS no: ACP-2021-521

We thank the reviewer for the constructive comments and suggestions that helped improve the manuscript. This document outlines the reviewer's comments (in **bold-blue**), followed by the author's responses and changes made in the revised manuscript (in *italics*). A marked-up version of the manuscript showing the revisions is appended to this response file.
* * *
**Comment:**

**Single scattering albedo (SSA) is a very important parameter in assessing the radiative impact of aerosols and on which there is meager data globally. The paper is a welcome addition to the aerosol literature in this regard. The authors have made use of satellite data of CERES and MODIS in obtaining global maps of SSA based on the concept of critical optical depth. They have presented the maps for different seasons as well considering a four year period. It is hoped such maps will be generated on annual basis subject to sufficient data availability.**

**The authors made a very clear presentation of the method of analysis including the error estimates. On the whole the paper will be a very important contribution to the area of aerosol radiative impact.**

We greatly appreciate and thank the reviewer for the summary evaluation, positive recommendations, and valuable feedback. Yes, on completion of peer-review, we intend to publish this dataset online and generate datasets for extended periods for future studies.

**Comment:**

**Specific comments/suggestions:**

1. **As described in the paper, the surface albedo is an important parameter in the SSA estimation. So the surface albedo maps for different seasons also should be given as in Fig 5 along with similar data for different seasons and regions in Table 1. This would greatly help in the discussion of the results.**

**Response:** This is indeed an insightful suggestion. We have added seasonal maps of surface albedo as Figure S2. And seasonal mean SSA values with standard deviation for each season have been listed in Table S2 for the various regions of interest.

**Additions to the supplementary file:**

[Figure]

**Figure S3.** Seasonal mean shortwave-integrated surface albedo from CERES

**Table S4**. Shortwave integrated seasonal mean surface albedo from CERES over regions of interest. Details of these regions are given in Table S1 and Fig. S1

| Region | Surface Albedo | | | |
|---|---|---|---|---|
| | DJF | MAM | JJA | SON |
| Canadian Boreal Forest | 0.36 ± 0.13 | 0.30 ± 0.12 | 0.12 ± 0.03 | 0.16 ± 0.05 |
| Russian Boreal Forest | 0.37 ± 0.10 | 0.27 ± 0.08 | 0.13 ± 0.02 | 0.20 ± 0.05 |
| South African Forest | 0.12 ± 0.01 | 0.13 ± 0.01 | 0.12 ± 0.02 | 0.13 ± 0.01 |
| Amazon Forest | 0.14 ± 0.01 | 0.14 ± 0.01 | 0.13 ± 0.02 | 0.14 ± 0.02 |
| North East Atlantic | 0.06 ± 0.01 | 0.05 ± 0.01 | 0.05 ± 0.01 | 0.05 ± 0.01 |
| South East Atlantic | 0.05 ± 0.01 | 0.05 ± 0.01 | 0.05 ± 0.01 | 0.05 ± 0.01 |
| Eastern Pacific | 0.05 ± 0.01 | 0.05 ± 0.00 | 0.05 ± 0.01 | 0.05 ± 0.00 |
| Sahara | 0.35 ± 0.06 | 0.34 ± 0.06 | 0.34 ± 0.06 | 0.34 ± 0.06 |
| Indo Gangetic Plain | 0.13 ± 0.02 | 0.13 ± 0.02 | 0.14 ± 0.02 | 0.13 ± 0.01 |
| Eastern China | 0.13 ± 0.04 | 0.13 ± 0.03 | 0.13 ± 0.03 | 0.13 ± 0.03 |

| | | | | |
|---|---|---|---|---|
| Arabian Sea | 0.06 ± 0.01 | 0.05 ± 0.01 | 0.05 ± 0.02 | 0.05 ± 0.01 |
| Bay of Bengal | 0.05 ± 0.01 | 0.05 ± 0.01 | 0.05 ± 0.01 | 0.05 ± 0.01 |

**Comment:**
**Specific comments/suggestions:**

2. **A brief description of the aerosol models used in the RT calculations should also be given.**

Thank you for the comment. We have included the aerosol model specifications in the supplementary file.

**Additions to the supplementary file:**

**Table S5:** Normalized extinction coefficient of the aerosol model

| λ (μm) | Ext$_{norm}$ | λ (μm) | Ext$_{norm}$ | λ (μm) | Ext$_{norm}$ |
|---|---|---|---|---|---|
| 0.25 | 1.597 | 0.75 | 0.847 | 3.2 | 0.5075 |
| 0.3 | 1.467 | 0.8 | 0.8202 | 3.39 | 0.5047 |
| 0.35 | 1.334 | 0.9 | 0.7828 | 3.5 | 0.5062 |
| 0.4 | 1.224 | 1 | 0.7536 | 3.75 | 0.4828 |
| 0.45 | 1.135 | 1.25 | 0.7038 | 4 | 0.4629 |
| 0.5 | 1.061 | 1.5 | 0.6706 | 4.5 | 0.4395 |
| 0.55 | 1 | 1.75 | 0.6349 | 5 | 0.4103 |
| 0.6 | 0.9505 | 2 | 0.5883 | | |
| 0.65 | 0.9106 | 2.5 | 0.4905 | | |
| 0.7 | 0.8757 | 3 | 0.491 | | |

**Table S6:** Phase function of the aerosol model (continued into Table S7)

| λ (μm) | Streams | | | | | | | |
|---|---|---|---|---|---|---|---|---|
| | 1 | 2 | 3 | 4 | 5 | 6 | 7 | 8 |
| **0.25** | 0.754 | 0.606 | 0.473 | 0.397 | 0.342 | 0.307 | 0.283 | 0.265 |
| **0.3** | 0.738 | 0.589 | 0.452 | 0.379 | 0.325 | 0.293 | 0.270 | 0.254 |
| **0.35** | 0.738 | 0.592 | 0.456 | 0.386 | 0.333 | 0.303 | 0.279 | 0.264 |
| **0.4** | 0.741 | 0.598 | 0.463 | 0.395 | 0.343 | 0.313 | 0.290 | 0.275 |
| **0.45** | 0.743 | 0.602 | 0.468 | 0.403 | 0.351 | 0.323 | 0.299 | 0.284 |
| **0.5** | 0.746 | 0.607 | 0.474 | 0.411 | 0.359 | 0.331 | 0.308 | 0.292 |
| **0.55** | 0.748 | 0.611 | 0.478 | 0.416 | 0.364 | 0.337 | 0.313 | 0.297 |
| **0.6** | 0.749 | 0.615 | 0.481 | 0.421 | 0.368 | 0.342 | 0.317 | 0.301 |
| **0.65** | 0.750 | 0.618 | 0.485 | 0.426 | 0.373 | 0.347 | 0.321 | 0.305 |
| **0.7** | 0.751 | 0.620 | 0.487 | 0.429 | 0.376 | 0.350 | 0.323 | 0.306 |
| **0.75** | 0.752 | 0.623 | 0.490 | 0.433 | 0.378 | 0.352 | 0.325 | 0.308 |
| **0.8** | 0.755 | 0.628 | 0.494 | 0.437 | 0.382 | 0.355 | 0.327 | 0.310 |
| **0.9** | 0.756 | 0.631 | 0.496 | 0.440 | 0.383 | 0.356 | 0.326 | 0.308 |
| **1** | 0.756 | 0.632 | 0.496 | 0.440 | 0.382 | 0.354 | 0.323 | 0.304 |
| **1.25** | 0.766 | 0.643 | 0.505 | 0.442 | 0.380 | 0.346 | 0.314 | 0.291 |

| | | | | | | | | |
|---|---|---|---|---|---|---|---|---|
| 1.5 | 0.777 | 0.651 | 0.512 | 0.441 | 0.376 | 0.337 | 0.302 | 0.276 |
| 1.75 | 0.798 | 0.673 | 0.536 | 0.455 | 0.385 | 0.339 | 0.300 | 0.271 |
| 2 | 0.826 | 0.707 | 0.577 | 0.491 | 0.415 | 0.362 | 0.316 | 0.282 |
| 2.5 | 0.858 | 0.750 | 0.636 | 0.552 | 0.476 | 0.418 | 0.365 | 0.323 |
| 3 | 0.871 | 0.765 | 0.662 | 0.578 | 0.505 | 0.444 | 0.391 | 0.346 |
| 3.2 | 0.836 | 0.708 | 0.584 | 0.491 | 0.414 | 0.354 | 0.304 | 0.264 |
| 3.39 | 0.818 | 0.682 | 0.548 | 0.453 | 0.375 | 0.317 | 0.270 | 0.233 |
| 3.5 | 0.808 | 0.670 | 0.530 | 0.434 | 0.356 | 0.299 | 0.253 | 0.217 |
| 3.75 | 0.805 | 0.667 | 0.524 | 0.429 | 0.349 | 0.292 | 0.246 | 0.210 |
| 4 | 0.797 | 0.660 | 0.513 | 0.421 | 0.340 | 0.284 | 0.238 | 0.202 |
| 4.5 | 0.795 | 0.655 | 0.507 | 0.413 | 0.331 | 0.275 | 0.228 | 0.192 |
| 5 | 0.808 | 0.663 | 0.520 | 0.420 | 0.338 | 0.278 | 0.230 | 0.192 |

**Table S7:** Phase function of aerosol model

| λ (μm) | Streams | | | | | | | |
|---|---|---|---|---|---|---|---|---|
| | 9 | 10 | 11 | 12 | 13 | 14 | 15 | 16 |
| 0.25 | 0.252 | 0.241 | 0.233 | 0.226 | 0.219 | 0.214 | 0.209 | 0.204 |
| 0.3 | 0.241 | 0.232 | 0.224 | 0.217 | 0.211 | 0.205 | 0.200 | 0.196 |
| 0.35 | 0.251 | 0.242 | 0.233 | 0.226 | 0.219 | 0.214 | 0.208 | 0.203 |
| 0.4 | 0.262 | 0.252 | 0.243 | 0.235 | 0.228 | 0.222 | 0.216 | 0.210 |
| 0.45 | 0.270 | 0.260 | 0.251 | 0.242 | 0.235 | 0.228 | 0.221 | 0.215 |
| 0.5 | 0.278 | 0.267 | 0.257 | 0.248 | 0.240 | 0.233 | 0.226 | 0.219 |
| 0.55 | 0.283 | 0.271 | 0.261 | 0.251 | 0.243 | 0.235 | 0.227 | 0.220 |
| 0.6 | 0.286 | 0.274 | 0.263 | 0.253 | 0.244 | 0.235 | 0.228 | 0.220 |
| 0.65 | 0.289 | 0.277 | 0.265 | 0.255 | 0.245 | 0.236 | 0.228 | 0.220 |
| 0.7 | 0.290 | 0.277 | 0.265 | 0.254 | 0.244 | 0.235 | 0.226 | 0.218 |
| 0.75 | 0.291 | 0.277 | 0.265 | 0.253 | 0.243 | 0.233 | 0.225 | 0.216 |
| 0.8 | 0.292 | 0.278 | 0.265 | 0.253 | 0.242 | 0.232 | 0.223 | 0.214 |
| 0.9 | 0.289 | 0.274 | 0.261 | 0.248 | 0.237 | 0.226 | 0.217 | 0.208 |
| 1 | 0.284 | 0.269 | 0.254 | 0.241 | 0.230 | 0.219 | 0.209 | 0.200 |
| 1.25 | 0.271 | 0.253 | 0.238 | 0.224 | 0.212 | 0.200 | 0.190 | 0.180 |
| 1.5 | 0.255 | 0.236 | 0.220 | 0.205 | 0.193 | 0.181 | 0.171 | 0.161 |
| 1.75 | 0.246 | 0.226 | 0.208 | 0.193 | 0.180 | 0.168 | 0.157 | 0.147 |
| 2 | 0.252 | 0.229 | 0.208 | 0.191 | 0.176 | 0.163 | 0.151 | 0.141 |
| 2.5 | 0.287 | 0.257 | 0.231 | 0.208 | 0.189 | 0.172 | 0.157 | 0.144 |
| 3 | 0.307 | 0.274 | 0.245 | 0.220 | 0.198 | 0.179 | 0.162 | 0.148 |
| 3.2 | 0.231 | 0.203 | 0.180 | 0.161 | 0.144 | 0.130 | 0.117 | 0.107 |
| 3.39 | 0.203 | 0.178 | 0.157 | 0.140 | 0.125 | 0.113 | 0.102 | 0.093 |
| 3.5 | 0.188 | 0.165 | 0.146 | 0.130 | 0.116 | 0.104 | 0.094 | 0.085 |
| 3.75 | 0.181 | 0.157 | 0.139 | 0.122 | 0.109 | 0.097 | 0.088 | 0.079 |
| 4 | 0.174 | 0.150 | 0.132 | 0.116 | 0.103 | 0.091 | 0.082 | 0.073 |
| 4.5 | 0.164 | 0.141 | 0.122 | 0.107 | 0.094 | 0.083 | 0.073 | 0.066 |
| 5 | 0.163 | 0.139 | 0.120 | 0.103 | 0.090 | 0.079 | 0.070 | 0.062 |

**Comment:**
Minor: Page 13, line 8 from top: the value is 0.83 and not 0.81 (Table 1)

Thank you for pointing this out. The table has now been shifted to supplementary file and the 'results and discussion' section is now completely revised based on reviewer #2's comments.

**Response to reviewer #2**

MS no: ACP-2021-521

We thank the reviewer for the constructive comments and suggestions that helped improve the manuscript. We have considered each comment carefully and revised the manuscript accordingly. This document outlines the reviewer's comments (in **bold-blue**), followed by the author's responses and changes made in the revised manuscript. A marked-up version of the manuscript showing the revisions is appended to this response file.
* * *
**Comment:**

**This paper combines daily CERES flux retrievals and MODIS aerosol retrievals to estimate aerosol single scattering albedo (SSA) at 550 nm. SSA is, after aerosol optical depth (AOD), the key parameter determining aerosols' radiative effect, but is difficult to retrieve well from most spaceborne measurements. The authors expand the application of a technique called "critical optical depth" they have developed before to a global scale. There is a brief comparison to airborne data, and to similar SSA maps available from OMI.**

**The study is in scope for the journal, though is also a close fit for AMT. It is fairly clearly presented. My main criticism is that the numerous uncertainties in the technique are glossed over and the reader is instead presented (in the abstract and conclusions) with the claim that the global uncertainty is about 0.03. There is no real analysis to back up this number and it seems to be based on limited airborne measurements over India and surrounding oceans. The manuscript is not very long and I think that the paper would benefit from a much more thorough and honest discussion and quantification of uncertainty sources. Otherwise an inexpert reader might believe the problem of determining SSA from space is essentially solved.**

**To that end, I recommend major revisions, and would like to review the revised version. I think the work is valuable but not yet at ACP quality.**

**Response:** We appreciate and thank the reviewer for the summary evaluation and valuable feedback. As suggested, we have included a more detailed uncertainty analysis and

comparisons with other datasets. These additional results, incorporated following the reviewer's comments, have vastly improved the manuscript.

**Comment:**

**The paper is missing references to the existing literature. For example, a lot of similar work has been done framed in terms of "critical reflectance" or "albedo" rather than "optical depth". The basic idea is the same, i.e. find a value of one parameter (surface/aerosol) where the top of atmosphere signal is invariant to changes in the other. Examples include Seidel and Popp (2012): https://amt.copernicus.org/articles/5/1653/2012/ and Wells et al (2012): https://doi.org/10.1029/2011JD016891 The authors should acknowledge and discuss the relative merits of other work using the same basic technique like this.**

**Response:** Comparison with Kaufman's critical reflectance method, which has a similar basic technique, is a valuable discussion. Thank you for bringing this point. The details and references to the existing literature on the critical reflectance method have been added. Their relative merits are also discussed.

**Additions to the revised manuscript:**

Page 2 Line 5 to 11: Fraser and Kaufman., 1985 developed a critical surface reflectance method to retrieve SSA using satellite data. Their method is based on radiative transfer simulations, which showed a particular surface reflectance for which the top of atmosphere albedo is independent of AOD. Upward radiances between a clear and a hazy day over a varying surface reflectance region are used, along with radiative transfer simulations, to derive SSA. This method has been widely applied to data from various satellites to derive SSA over particular regions (Kaufman, 1987; Kaufman et al., 1990, 2001; Zhu et al., 2011; Wells et al., 2012). Seidel and Popp., 2012 have done extensive studies on the method's sensitivity to various parameters.

Page 2 Line 24 to Page 3 Line 2: The "critical optical depth" method developed in this research paper shares a similar concept to the critical surface reflectance method (Fraser and Kaufman., 1985). For a particular parameter (such as surface reflectance or optical depth), there exists a critical value at which the top of atmosphere albedo can be considered independent of variations in that parameter. Both the methods retrieve SSA by parameterizing the critical value as a function of SSA using radiative transfer

simulations. The critical reflectance method requires two-days data and large variations in surface reflectance over the region. It's suitable for retrieving daily SSA for a particular region. Whereas the critical optical method developed in this paper is suitable for retrieving monthly or seasonal global maps of SSA.

**References added**

Kaufman, Y. J., Fraser, R. S., and Ferrare, R. A.: Satellite measurements of large-scale air pollution: methods, J. Geophys. Res., 95, 9895–9909, https://doi.org/10.1029/JD095iD07p09895, 1990.

Kaufman, Y. J., Tanré, D., Dubovik, O., Karnieli, A., and Remer, L. A.: Absorption of sunlight by dust as inferred from satellite and ground-based remote sensing, Geophys. Res. Lett., 28, 1479–1482, https://doi.org/10.1029/2000GL012647, 2001.

Seidel, F. C. and Popp, C.: Critical surface albedo and its implications to aerosol remote sensing, Atmos. Meas. Tech., 5, 1653–1665, https://doi.org/10.5194/AMT-5-1653-2012, 2012.

Wells, K. C., Martins, J. V., Remer, L. A., Kreidenweis, S. M., and Stephens, G. L.: Critical reflectance derived from MODIS: Application for the retrieval of aerosol absorption over desert regions, J. Geophys. Res. Atmos., 117, 3202, https://doi.org/10.1029/2011JD016891, 2012.

**Comment:**
The uncertainty discussion really needs to be strengthened. There are claims of 0.03 throughout the paper but they are really not supported. The authors do not really acknowledge that e.g. aerosol vertical location matters a lot as well: you can get quite a different forcing if the aerosols change height due to interactions with Rayleigh scattering (which depends on pressure). This is well established by e.g. the OMI and combined UV-vis work the authors cite during the paper. Other key uncertainty sources are inconsistencies between the aerosol and surface properties assumed by the MODIS and CERES retrievals with each other and with the OPAC-based SBDART calculations. A further is the possibility of variation on scales of the regions used for the linear fitting process; the residuals on the fit (uncertainty on the intercept) would be one easy way to incorporate this effect. There are doubtless others as well. I think the authors need to list the potential uncertainty sources and try to quantify as many as possible – even if approximately – so it becomes clear which are the most important. The comparison

**against airborne data is good to have but this is only a small part of the picture, and definitely not enough by itself.**

**Response:** Thank you for this suggestion on strengthening the uncertainty discussion. We have included a new section on uncertainty analysis (section 5) where retrieval uncertainties due to possible perturbations in various parameters have been calculated and presented.

**Additions to the revised manuscript:**

**5 Uncertainty Analysis**

Table 1 identifies the major sources of error in the retrieval and summarizes their individual contribution. Uncertainty in the retrieved SSA was estimated by calculating retrieval sensitivities to perturbations in the possible error sources. The range of perturbation was based on published literature or reasonable assumptions for possible variations.

**Table 1.** Estimates of the uncertainty in retrieved SSA

| Parameter | Input Uncertainty | Retrieval Uncertainty |
|---|---|---|
| Surface albedo | ±0.01 | ±0.03 |
| AOD | 20% ±0.05 (land) 5% ±0.03 (ocean) | ±0.02 |
| Angstrom exponent | ±0.4 | ±0.01 |
| Refractive index | ±0.01 | ±0.01 |
| Aerosol height | ±1 km | ±0.01 |
| Aerosol type | Smoke vs dust | ±0.01 |
| Residual of fit | ±0.05 | ±0.02 |

Uncertainty in shortwave integrated surface albedo from CERES results in the maximum uncertainty in SSA of ±0.03. MODIS retrieved aerosol optical depth contains considerable uncertainties due to assumed aerosol models (Jeong et al., 2005). The MODIS aerosol optical depth uncertainty is 20% ±0.05 over land (Chu et al., 2002) and 5% ±0.03 over the ocean (Remer et al., 2002). The corresponding error in our retrieval is ±0.02. For a typical variation of angstrom exponent (±0.4) and refractive index

($\pm0.01$), the uncertainties vary depending on the surface albedo and are mostly around $\pm0.01$.

Changes in aerosol height can vary the TOA radiances due to Rayleigh scattering interactions, which depend on pressure. Sensitivity to aerosol height was estimated by conducting a synthetic retrieval of SSA over a range of aerosol height values and perturbations from those heights. The average uncertainty observed for an aerosol height variation of $\pm1$ km was $\pm0.01$. Many methods have been developed for detecting aerosol type, especially smoke vs. dust, to improve the uncertainties of various AOD and SSA retrievals.

Uncertainties due to possible variations on scales of the regions used for linear fitting were estimated as residuals of the fit. The uncertainty on the linear intercept is spatially dependent and is mostly around $\pm0.02$, with higher values for those combinations having a slope close to zero during the regression. For highly correlated cases (i.e., correlation coefficient $|r| > 0.5$), the probability of obtaining a slope close to zero is ~20% over the ocean and <5% over land. These cases are mostly formed over regions where AOD variations are less. Regions having large variations in AOD values have lower uncertainty due to residual fit.

Overall, the algorithm is most sensitive to variations in surface albedo, followed by higher sensitivity towards AOD values used in the linear fit. The uncertainties are higher for scattering aerosols over bright surfaces and absorbing aerosols above dark surfaces. Sensitivity to water vapor is almost negligible, except in very few cases where the uncertainty is $\pm 0.008$. The CERES-MODIS algorithm is most effective over regions with large AOD variations and less surface albedo variations.

**Comment:**
**It is not clear to me if the derived SSA data are publicly available; I did not see a link in the paper. They ideally should be somewhere.**

**Response:** These datasets were generated as part of the author's ongoing Ph.D. research. All the datasets generated as part of the thesis work will be published online on the department's website on successful completion of the degree. For now, as mentioned in the data availability section, it will be available on request.

**Comment:**
**Specific comments:**

**1. Abstract: As SSA is a spectrally varying quantity, the wavelength reported should be given here (550 nm). Additionally, the statement about uncertainty is basically unsupported and seems to come from the comparison against a small number of aircraft observations. I recommend that this statement is removed or made a lot weaker, e.g. "limited comparisons against airborne observations over India and surrounding oceans were generally in agreement within +-0.03". The abstract should be an honest summary of what is in the paper, not a place to hype up the work, as unfortunately many people only read abstracts and skim papers.**

**Response:** We agree with the comment. The wavelength of retrieved SSA (550 nm) is now mentioned in the abstract Line 3. And the two sentences related to aircraft data and uncertainty have been replaced with the recommended statement, "Limited comparisons against airborne observations over India and surrounding oceans were generally within ±0.03."

**Comment:**
**2. Page 6 line 15: what exactly is the significance test done on here? This should be clearer. My guess is that it is on the linear correlation coefficient between the AOD and albedo difference, i.e. the authors are testing whether the probability of observing a correlation coefficient at least that large if there were truly no linear relationship between the two quantities is 0.05 or lower. Is that right?**

**Response:** Yes, we are testing the significance of the correlation coefficient calculated during the linear regression between AOD and ΔAlbedo. The significance level is taken as 0.05, indicating that the risk of concluding that a correlation exists- when actually no correlation exists- is less than or equal to 5%. We have rephrased the sentence to improve the clarity.

> **Additions to revised manuscript:**
>
> Page 7 Line 15 to 17: A significance test on the correlation coefficient between AOD and ΔAlbedo is performed with a 0.05 significance level. Only those τc values obtained through regressions that are statistically significant at 95% confidence level are utilized further to retrieve SSA.

**Comment:**
**3. Figure 2: here and in the text, it is mentioned that small regression slopes mean that SSA cannot be determined well. Again, the linear model fit uncertainties (see general**

comments) could be used to do this for every grid cell and associate an uncertainty. If this is not done, though, then the authors should show where and how frequently these conditions occur. It is not clear if, for example, if almost never happens or is common. In the latter case the seasonal maps shown later may have some additional sampling-related uncertainties.

**Response:** Following the reviewer's general comment #1, this point has been clarified in the revised manuscript's new Section 5 Uncertainty analysis.

**Additions to revised manuscript:**

Page 13 Line 12 to 17: Uncertainties due to possible variations on scales of the regions used for linear fitting were estimated as residuals of the fit. The uncertainty on the linear intercept is spatially dependent and is mostly around ±0.02, with higher values for those combinations having a slope close to zero during the regression. For highly correlated cases (i.e., correlation coefficient $| r | > 0.5$), the probability of obtaining a slope close to zero is ~20% over the ocean and <5% over land. These cases are mostly formed over regions where AOD variations are less. Regions having large variations in AOD values have lower uncertainty due to residual fit.

**Comment:**
4.    Section 4: more information about the airborne measurement of SSA, including their uncertainty, is necessary. I encourage the authors to search for additional airborne data which may supplement their results from elsewhere in the world, which would strengthen the robustness of these comparisons. There have for example been NASA field campaigns through the US, south-eastern Atlantic, and Korea during this time frame, and these NASA data are publicly available (I am sure the investigators who spent considerable time collecting the data would be glad to see them used). Doubtless there are other resources as well.

**Response:** Additional information about the aircraft measurements and their uncertainties has been included in the revised manuscript.

**Additions to the revised manuscript (**shown in black font color**):**

Page 14 Line 1 to 13: Babu et al. (2016), as part of RAWEX (Moorthy et al., 2016), derived SSA at 520 nm from aircraft measurements of scattering and absorption coefficients over the Indo-Gangetic Plain (IGP) and Central India during winter 2012

and spring/pre-monsoon 2013. Various measurements of aerosol properties were carried out in an instrumented Beechcraft B200 aircraft of the National Remote Sensing Centre, India. Manoj et al. (2019) estimated vertical profiles of SSA during the SWAAMI campaign conducted during monsoon (June - July) 2016 over IGP, Arabian Sea, and Bay of Bengal. Aerosol scattering coefficients were measured aboard the Facility for Airborne Atmospheric Measurements (FAAM) BAe-146 aircraft. Vaishya et al. (2018) estimated vertical profiles of SSA (520 nm) using an instrumented aircraft, Beechcraft B200, during SWAAMI-RAWEX campaign (June 2016). Instrument design and calibration were based on Anderson et al., 1996 and its application for Indian field experiments was as described by Nair et al., 2009. Uncertainties in the scattering coefficient measurement by nephelometer are ~±10%, as reported by Anderson et al., 1996. As stated by Babu et al., 2016 uncertainties in the columnar SSA values estimated from RAWEX aircraft measurements depend mainly on instrumental uncertainties, sampling errors, and large spatial averaging.

**Response continued:**

Initially, while making the aircraft data comparisons, we had looked into other aircraft data available from various field campaigns such as ORACLES (South Eastern Atlantic), ACE-ENA (Northeastern Atlantic), DISCOVER-AQ (USA), etc. These flight datasets are available in the ASDC and ESPO data archives. They provide the raw data collected during the flights – such scattering coefficient measured by nephelometer at various latitude, longitude, and altitudes. These raw datasets need to be carefully processed considering the various instrument calibrations and experimental setup to obtain the scattering coefficient profiles over the flight track, from which the SSA profiles are computed. Further, these profiles need to be vertically integrated (also considering the flight's lat-lon variations) to obtain the columnar SSA required for comparison with the CERES-MODIS dataset. This entire work in itself would be an extensive experimental-data processing of the flight data. Carrying out these detailed computations would only provide a few datapoints for comparison with CERES-MODIS values over that region for the period.

The SWAAMI, RAWEX, and SWAAMI-RAWEX campaigns were organized and conducted by our research group. Hence the processed-datasets generated from the raw data by the experimentalists were readily available to us for comparison with the CERES-MODIS satellite data.

The suggestion provided by the reviewer to include other publicly available aircraft datasets is really a valuable point. It would surely add more points to the aircraft comparisons. But since this paper focuses more on satellite data and algorithms, performing extensive experimental calculations to obtain just a few data points would be tedious.

Instead, following reviewer's comment #6, we have included comparisons with POLDER and AERONET sites. AERONET sites were chosen based on the classification provided in Giles et al., 2012. The reviewer's suggestions to include these other datasets have significantly improved the manuscript. With these additional results now included in the revised manuscript, the addition of a few data points obtained from extensive flight data computations may not make significant improvements.

**Comment:**
**5.    Section 4: I disagree with the framing of this section as a "validation" given the small extent of comparison and lack of detail or consideration of uncertainties. I suggest that it be renamed "Comparison with airborne observations" and the use of the word "validation" throughout be changed.**

**Response:** Done. Section 4 title has been rephrased as "Comparison with airborne observations." All usage of 'validation' with aircraft data has been replaced with 'comparison' throughout the revised manuscript.

**Comment:**
**6.    Section 5: comparing against OMI is one good choice; the authors might also mention the POLDER archive, which is similar or higher quality for SSA, but ended in 2013 before the time period the authors used here. The results could also be compared to global aerosol model simulations or reanalyses. And, although the authors briefly mention AERONET, it would be worthwhile to add a comparison with AERONET for regions where there is a persistent repeatable high aerosol loading. The authors could take AERONET climatologies themselves or go to other analyses, e.g. Giles et al (2012) https://doi.org/10.1029/2012JD018127 for various types of aerosol or Sayer et al (2014) https://acp.copernicus.org/articles/14/11493/2014/ for smoke in various regions. POLDER results could also be used in a climatological sense. Finally here the authors should be clearer that the reason for less OMI coverage over oceans is not so much cloudiness but in fact that over ocean the OMI retrieval is only done if the UVAI is high (I believe 0.7 or above). So this introduces a sampling bias towards high-AOD, high**

**absorption cases (as we know baseline sea spray is not very absorbing) which is likely the main reason that OMI SSA is patchier and has lower values over ocean. This could be tested by also subsampling the MODIS-CERES data to only examine those times when OMI also has a retrieval. POLDER and reanalyses do not have this issue. Expanding the comparisons would provide further evidence for where the authors' technique may be valuable or where there are issues with one or other data set.**

**Response:** Thank you for these suggestions to include comparisons with other satellite and ground-based SSA datasets. These suggestions have helped improve the revised manuscript. Summary of work done based on this comment:

- Alongside OMI SSA (500 nm), we have compared the CERES-MODIS SSA dataset (550 nm) with POLDER climatological SSA (565 nm).

- The complete "results and discussion" section has been rewritten, emphasizing the advantages of each of the three datasets and the issues with one or the other data set. The table containing seasonal SSA values for different regions of interest has been shifted to the supplementary file.

- For the study period of 2014-18, the CERES-MODIS SSA has also been compared to monthly AERONET SSA data (440 nm) for the corresponding period for various AERONET sites. As suggested by the reviewer, we have chosen AERONET sites based on the type of aerosols as given by Giles et al., 2012. These results have been incorporated as the new Section 7 in the revised manuscript.

- Following the reviewer's comment, we intended to compare with the MERRAero reanalysis dataset. It would have been a valuable addition to the paper. Unfortunately, the OpenDap server was not accessible for downloading the data. Monthly SSA data files downloaded from the GEOS-5 data server were missing data values in it. We tried with the GrADS data server, but the download link was unavailable. We had also got our manuscript response deadline extended with ACP, hoping the data server would be up running by then. But we couldn't get to download the files.

**Additions to the revised manuscript:**

[revised manuscript text omitted]

**Comment:**

**In summary, this study has value but I think a much deeper treatment of uncertainty is needed. Otherwise it is not clear to what extent this technique improves our understanding of aerosol SSA, or where the largest challenges remain. We can't quantitatively move forward if we don't understand where we stand now.**

**Response:** Uncertainty analysis has been studied for various parameters and is now presented in a separate section 5. CERES-MODIS dataset has been compared both with OMI and POLDER. Their respective advantages and suitable area of usage are also discussed in the section 4, 'results and discussion'. Comparison were done with monthly AERONET data from 15 sites, chosen based on Giles et al., 2012 (section 7). We thank the reviewer for these detailed suggestions and comments, which greatly improved the manuscript.

[revised manuscript text omitted]
 | 0.91 ± 0.01 (0.92 ± 0.01) [0.93 ± 0.02] | 0.90 ± 0.01 (0.94 ± 0.01) [0.91 ± 0.02] | 0.91 ± 0.02 (0.95 ± 0.01) [0.95 ± 0.02] | 0.91 ± 0.02 (0.94 ± 0.02) [0.93 ± 0.03] |

Formatted Table

[Figure]

**Figure S2.** Map showing location of AERONET sites used in this study. The type of aerosols (dust, mixed, urban and biomass) were as defined in Giles et al., 2012

Table S3: Name of AERONET site as shown in Fig. S2

| No. | Name | No. | Name | No. | Name |
|---|---|---|---|---|---|
| 1 | GSFC | 6 | Capo_Verde | 11 | SEDE_BOKER |
| 2 | Mexico_City | 7 | Dakar | 12 | Kanpur |
| 3 | Alta_Floresta | 8 | Illorin | 13 | XiangHe |
| 4 | Ispra | 9 | Banizoumbou | 14 | Shirahama |
| 5 | Moldova | 10 | Mongu | 15 | Lake_Argyle |

[Figure]

**Figure S3.** Seasonal mean shortwave-integrated surface albedo from CERES

**Table S4**. Shortwave integrated seasonal mean surface albedo from CERES over regions of interest. Details of these regions are given in Table S1 and Fig. S1

| Region | Surface Albedo | | | |
|---|---|---|---|---|
| | DJF | MAM | JJA | SON |
| Canadian Boreal Forest | 0.36 ± 0.13 | 0.30 ± 0.12 | 0.12 ± 0.03 | 0.16 ± 0.05 |
| Russian Boreal Forest | 0.37 ± 0.10 | 0.27 ± 0.08 | 0.13 ± 0.02 | 0.20 ± 0.05 |
| South African Forest | 0.12 ± 0.01 | 0.13 ± 0.01 | 0.12 ± 0.02 | 0.13 ± 0.01 |
| Amazon Forest | 0.14 ± 0.01 | 0.14 ± 0.01 | 0.13 ± 0.02 | 0.14 ± 0.02 |
| North East Atlantic | 0.06 ± 0.01 | 0.05 ± 0.01 | 0.05 ± 0.01 | 0.05 ± 0.01 |
| South East Atlantic | 0.05 ± 0.01 | 0.05 ± 0.01 | 0.05 ± 0.01 | 0.05 ± 0.01 |
| Eastern Pacific | 0.05 ± 0.01 | 0.05 ± 0.00 | 0.05 ± 0.01 | 0.05 ± 0.00 |
| Sahara | 0.35 ± 0.06 | 0.34 ± 0.06 | 0.34 ± 0.06 | 0.34 ± 0.06 |
| Indo Gangetic Plain | 0.13 ± 0.02 | 0.13 ± 0.02 | 0.14 ± 0.02 | 0.13 ± 0.01 |
| Eastern China | 0.13 ± 0.04 | 0.13 ± 0.03 | 0.13 ± 0.03 | 0.13 ± 0.03 |
| Arabian Sea | 0.06 ± 0.01 | 0.05 ± 0.01 | 0.05 ± 0.02 | 0.05 ± 0.01 |
| Bay of Bengal | 0.05 ± 0.01 | 0.05 ± 0.01 | 0.05 ± 0.01 | 0.05 ± 0.01 |

**Formatted Table**

**Table S5:** Normalized extinction coefficient of the aerosol model

| λ (μm) | Ext$_{norm}$ | λ (μm) | Ext$_{norm}$ | λ (μm) | Ext$_{norm}$ |
|--------|--------------|--------|--------------|--------|--------------|
| 0.25 | 1.597 | 0.75 | 0.847 | 3.2 | 0.5075 |
| 0.3 | 1.467 | 0.8 | 0.8202 | 3.39 | 0.5047 |
| 0.35 | 1.334 | 0.9 | 0.7828 | 3.5 | 0.5062 |
| 0.4 | 1.224 | 1 | 0.7536 | 3.75 | 0.4828 |
| 0.45 | 1.135 | 1.25 | 0.7038 | 4 | 0.4629 |
| 0.5 | 1.061 | 1.5 | 0.6706 | 4.5 | 0.4395 |
| 0.55 | 1 | 1.75 | 0.6349 | 5 | 0.4103 |
| 0.6 | 0.9505 | 2 | 0.5883 | | |
| 0.65 | 0.9106 | 2.5 | 0.4905 | | |
| 0.7 | 0.8757 | 3 | 0.491 | | |

**Table S6:** Phase function of the aerosol model  (continued into Table S7)

| λ (μm) | Streams | | | | | | | |
|--------|-------|-------|-------|-------|-------|-------|-------|-------|
| | 1 | 2 | 3 | 4 | 5 | 6 | 7 | 8 |
| 0.25 | 0.754 | 0.606 | 0.473 | 0.397 | 0.342 | 0.307 | 0.283 | 0.265 |
| 0.3 | 0.738 | 0.589 | 0.452 | 0.379 | 0.325 | 0.293 | 0.270 | 0.254 |
| 0.35 | 0.738 | 0.592 | 0.456 | 0.386 | 0.333 | 0.303 | 0.279 | 0.264 |
| 0.4 | 0.741 | 0.598 | 0.463 | 0.395 | 0.343 | 0.313 | 0.290 | 0.275 |
| 0.45 | 0.743 | 0.602 | 0.468 | 0.403 | 0.351 | 0.323 | 0.299 | 0.284 |
| 0.5 | 0.746 | 0.607 | 0.474 | 0.411 | 0.359 | 0.331 | 0.308 | 0.292 |
| 0.55 | 0.748 | 0.611 | 0.478 | 0.416 | 0.364 | 0.337 | 0.313 | 0.297 |
| 0.6 | 0.749 | 0.615 | 0.481 | 0.421 | 0.368 | 0.342 | 0.317 | 0.301 |
| 0.65 | 0.750 | 0.618 | 0.485 | 0.426 | 0.373 | 0.347 | 0.321 | 0.305 |
| 0.7 | 0.751 | 0.620 | 0.487 | 0.429 | 0.376 | 0.350 | 0.323 | 0.306 |
| 0.75 | 0.752 | 0.623 | 0.490 | 0.433 | 0.378 | 0.352 | 0.325 | 0.308 |
| 0.8 | 0.755 | 0.628 | 0.494 | 0.437 | 0.382 | 0.355 | 0.327 | 0.310 |
| 0.9 | 0.756 | 0.631 | 0.496 | 0.440 | 0.383 | 0.356 | 0.326 | 0.308 |
| 1 | 0.756 | 0.632 | 0.496 | 0.440 | 0.382 | 0.354 | 0.323 | 0.304 |
| 1.25 | 0.766 | 0.643 | 0.505 | 0.442 | 0.380 | 0.346 | 0.314 | 0.291 |
| 1.5 | 0.777 | 0.651 | 0.512 | 0.441 | 0.376 | 0.337 | 0.302 | 0.276 |
| 1.75 | 0.798 | 0.673 | 0.536 | 0.455 | 0.385 | 0.339 | 0.300 | 0.271 |
| 2 | 0.826 | 0.707 | 0.577 | 0.491 | 0.415 | 0.362 | 0.316 | 0.282 |
| 2.5 | 0.858 | 0.750 | 0.636 | 0.552 | 0.476 | 0.418 | 0.365 | 0.323 |
| 3 | 0.871 | 0.765 | 0.662 | 0.578 | 0.505 | 0.444 | 0.391 | 0.346 |
| 3.2 | 0.836 | 0.708 | 0.584 | 0.491 | 0.414 | 0.354 | 0.304 | 0.264 |
| 3.39 | 0.818 | 0.682 | 0.548 | 0.453 | 0.375 | 0.317 | 0.270 | 0.233 |
| 3.5 | 0.808 | 0.670 | 0.530 | 0.434 | 0.356 | 0.299 | 0.253 | 0.217 |
| 3.75 | 0.805 | 0.667 | 0.524 | 0.429 | 0.349 | 0.292 | 0.246 | 0.210 |
| 4 | 0.797 | 0.660 | 0.513 | 0.421 | 0.340 | 0.284 | 0.238 | 0.202 |
| 4.5 | 0.795 | 0.655 | 0.507 | 0.413 | 0.331 | 0.275 | 0.228 | 0.192 |
| 5 | 0.808 | 0.663 | 0.520 | 0.420 | 0.338 | 0.278 | 0.230 | 0.192 |

**Table S7:** Phase function of aerosol model

| λ (μm) | Streams | | | | | | | |
|---|---|---|---|---|---|---|---|---|
| | 9 | 10 | 11 | 12 | 13 | 14 | 15 | 16 |
| 0.25 | 0.252 | 0.241 | 0.233 | 0.226 | 0.219 | 0.214 | 0.209 | 0.204 |
| 0.3 | 0.241 | 0.232 | 0.224 | 0.217 | 0.211 | 0.205 | 0.200 | 0.196 |
| 0.35 | 0.251 | 0.242 | 0.233 | 0.226 | 0.219 | 0.214 | 0.208 | 0.203 |
| 0.4 | 0.262 | 0.252 | 0.243 | 0.235 | 0.228 | 0.222 | 0.216 | 0.210 |
| 0.45 | 0.270 | 0.260 | 0.251 | 0.242 | 0.235 | 0.228 | 0.221 | 0.215 |
| 0.5 | 0.278 | 0.267 | 0.257 | 0.248 | 0.240 | 0.233 | 0.226 | 0.219 |
| 0.55 | 0.283 | 0.271 | 0.261 | 0.251 | 0.243 | 0.235 | 0.227 | 0.220 |
| 0.6 | 0.286 | 0.274 | 0.263 | 0.253 | 0.244 | 0.235 | 0.228 | 0.220 |
| 0.65 | 0.289 | 0.277 | 0.265 | 0.255 | 0.245 | 0.236 | 0.228 | 0.220 |
| 0.7 | 0.290 | 0.277 | 0.265 | 0.254 | 0.244 | 0.235 | 0.226 | 0.218 |
| 0.75 | 0.291 | 0.277 | 0.265 | 0.253 | 0.243 | 0.233 | 0.225 | 0.216 |
| 0.8 | 0.292 | 0.278 | 0.265 | 0.253 | 0.242 | 0.232 | 0.223 | 0.214 |
| 0.9 | 0.289 | 0.274 | 0.261 | 0.248 | 0.237 | 0.226 | 0.217 | 0.208 |
| 1 | 0.284 | 0.269 | 0.254 | 0.241 | 0.230 | 0.219 | 0.209 | 0.200 |
| 1.25 | 0.271 | 0.253 | 0.238 | 0.224 | 0.212 | 0.200 | 0.190 | 0.180 |
| 1.5 | 0.255 | 0.236 | 0.220 | 0.205 | 0.193 | 0.181 | 0.171 | 0.161 |
| 1.75 | 0.246 | 0.226 | 0.208 | 0.193 | 0.180 | 0.168 | 0.157 | 0.147 |
| 2 | 0.252 | 0.229 | 0.208 | 0.191 | 0.176 | 0.163 | 0.151 | 0.141 |
| 2.5 | 0.287 | 0.257 | 0.231 | 0.208 | 0.189 | 0.172 | 0.157 | 0.144 |
| 3 | 0.307 | 0.274 | 0.245 | 0.220 | 0.198 | 0.179 | 0.162 | 0.148 |
| 3.2 | 0.231 | 0.203 | 0.180 | 0.161 | 0.144 | 0.130 | 0.117 | 0.107 |
| 3.39 | 0.203 | 0.178 | 0.157 | 0.140 | 0.125 | 0.113 | 0.102 | 0.093 |
| 3.5 | 0.188 | 0.165 | 0.146 | 0.130 | 0.116 | 0.104 | 0.094 | 0.085 |
| 3.75 | 0.181 | 0.157 | 0.139 | 0.122 | 0.109 | 0.097 | 0.088 | 0.079 |
| 4 | 0.174 | 0.150 | 0.132 | 0.116 | 0.103 | 0.091 | 0.082 | 0.073 |
| 4.5 | 0.164 | 0.141 | 0.122 | 0.107 | 0.094 | 0.083 | 0.073 | 0.066 |
| 5 | 0.163 | 0.139 | 0.120 | 0.103 | 0.090 | 0.079 | 0.070 | 0.062 |

**Response to reviewer #2's reply**

MS no: ACP-2021-521

We thank the reviewer for the constructive feedback and suggestions. We have considered each comment carefully and revised the manuscript accordingly. This document outlines the reviewer's comments (in **bold-blue**), followed by the author's responses and changes made in the revised manuscript. A marked-up version of the manuscript showing the revisions is appended to this response file.
* * *
**Comment:**

**I reviewed the previous version of this manuscript, and recommended major revisions. In this version the authors have made numerous revisions, including additional analyses and various clarifications. These have strengthened the manuscript, and I appreciate the authors' efforts. The reason for not including additional aircraft information makes sense.**

**The paper is better but some aspects needs further clarification and possibly correction. Because of this, I recommend further revisions. I would be happy to review again if the Editor feels this would be useful. I expect the next revision is likely to be the final one before acceptance. It is an interesting study, just is still not quite up to ACP standards in my opinion. My comments on this version are as follows:**

**Response:** We thank the reviewer for the positive comments and valuable feedback. As suggested, we have included more clarifications and additional information. These added results, we believe, have improved the manuscript further.

**Comment:**

**1. Page 2 line 8: I believe this should say top of atmosphere reflectance rather than albedo. This and the other applications (e.g. Wells, Seidel/Popp) used directional (radiance/reflectance) data rather than albedo. The authors should check the use of reflectance vs. albedo throughout as some instances refer to directional measurements (e.g. satellite measurements) while others are directionally integrated (e.g. CERES flux retrievals).**

**Response:** Thank you for pointing out this difference between the TOA reflectance used in literature and the TOA albedo used in our work. We have gone through, checked, and corrected all instances of usage of reflectance and albedo in the manuscript.

**Comment:**
**2. Looking at the revised paper and Supplement, I still have questions about the aerosol model used for SSA retrieval (section 3.2). That refers to tables S5, S6, and S7. Section 3.2 says "aerosol models from OPAC" (models being plural) but the Supplement seems to describe only one optical model. Is the same model used globally? Table S5 makes sense, but I do not think tables S6 and S7 are really phase function values (those would be a function of angle). I am not sure what 'streams' refers to here and am guessing these might be related to SBDART's angular decomposition for the discrete ordinate method? In any case I would just give spectral extinction (as provided in Table S5) together with size distribution and refractive index (as well as perhaps further derived parameters like spectral SSA and asymmetry parameter) rather than providing spectral angular phase function. Or is perhaps the same spectral extinction used everywhere but SSA is manually varied in the calculation (in which case is e.g. phase function also varied)? These aspects remain unclear. So, as well as rewording section 3.2 to clarify if this is one model (manually varied SSAs) or multiple models (e.g. different size distributions and refractive indices forward propagated through Mie code), these tables should be corrected. The spectral dependence of Table S5 suggests to me that this is a fairly mixed fine/coarse aerosol model which implies potentially larger errors for fine-dominated or coarse-dominated aerosol plumes, but again it is not quite clear what is done.**

**Response:** OPAC provides the user with facilities to generate their own set of aerosol models, as well as a set of default aerosol models (such as the clean ocean, polluted ocean, arid, clean land, polluted land, and highly polluted land). Details of these models are documented in OPAC literature (Hess et al., 1998); hence we had mentioned only the details of one of the models in the last revised manuscript. Also, since the default sets of models were used, not much emphasis was given to aerosol types in the manuscript.

We completely agree that these points need further clarification. The following details, tables, and figures have been added to the supplementary file and mentioned in the manuscript:

**Additions to supplementary file:**

**Details of aerosol models used**

The aerosol models used are from OPAC (Optical Properties of Aerosols and Clouds), developed by Hess et al., (1998). The existing mixture of aerosol types in OPAC is used – clean ocean, polluted ocean, arid, clean land, polluted land, and highly polluted land.

A LUT is indexed by surface albedo, water vapour, and SSA. The LUT of the aerosol type selected for the pixel is used to compute SSA from $\tau_c$.

[Figure]

**Fig S4.** An inverse look-up is performed to computer SSA from $\tau_c$. Details of the "Identify Aerosol Model" block are shown in Fig S5.

The aerosol type is selected based on geographical location (Ocean/land, surface albedo) and aerosol loading (AOD).

[Figure]

**Fig S5.** (a) Decision tree for selecting the aerosol model. (b) Shows a sample map of the aerosol model used for a particular day 03 Jun 2017, following the color code used in the decision tree

**Table S5.** Components of the mixed aerosol types used

| Aerosol Type | Components | Number density (1/cm^3) | Number mixing ratio | Volume mixing ratio | Mass mixing ratio |
|---|---|---|---|---|---|
| clean ocean | waso | 1.50E+03 | 9.87E-01 | 6.44E-02 | 7.05E-02 |
| | ssam | 2.00E+01 | 1.32E-02 | 9.15E-01 | 9.09E-01 |
| | sscm | 3.20E-03 | 2.11E-06 | 2.03E-02 | 2.01E-02 |
| Polluted ocean | waso | 3.80E+03 | 4.22E-01 | 1.47E-01 | 1.60E-01 |
| | soot | 5.18E+03 | 5.76E-01 | 8.53E-03 | 6.53E-03 |
| | ssam | 2.00E+01 | 2.22E-03 | 8.26E-01 | 8.15E-01 |
| | sscm | 3.20E-03 | 3.56E-07 | 1.83E-02 | 1.80E-02 |
| Arid | waso | 2.00E+03 | 8.70E-01 | 3.19E-02 | 1.77E-02 |
| | minm | 2.70E+02 | 1.17E-01 | 3.26E-02 | 3.31E-02 |
| | miam | 3.05E+01 | 1.33E-02 | 7.35E-01 | 7.46E-01 |
| | micm | 1.42E-01 | 6.17E-05 | 2.00E-01 | 2.03E-01 |
| Clean land | inso | 1.50E-01 | 5.77E-05 | 3.27E-01 | 4.07E-01 |
| | waso | 2.60E+03 | 1.00E+00 | 6.73E-01 | 5.93E-01 |
| Polluted land | inso | 4.00E-01 | 2.61E-05 | 3.15E-01 | 3.96E-01 |
| | waso | 7.00E+03 | 4.58E-01 | 6.53E-01 | 5.83E-01 |
| | soot | 8.30E+03 | 5.43E-01 | 3.29E-02 | 2.07E-02 |
| Highly polluted land | inso | 6.00E-01 | 1.20E-05 | 2.28E-01 | 2.99E-01 |
| | waso | 1.57E+04 | 3.14E-01 | 7.07E-01 | 6.58E-01 |
| | soot | 3.43E+04 | 6.86E-01 | 6.56E-02 | 4.31E-02 |

**

*inso – insoluble*                    *minm – mineral (nuclei mode)*
*waso – water soluble*             *miam – mineral (accumulation mode)*
*ssam – sea salt (accumulation mode)*   *micm – mineral (coarse mode)*
*sscm - sea salt (coarse mode)*

*\* More details such as refractive index and size distributions can be referred to in Hess et al 1998*

**Table S6.** Normalized extinction coefficient for each aerosol type

| Aerosol Type | Wavelength (microns) | | | | | | | | | | | |
|---|---|---|---|---|---|---|---|---|---|---|---|---|
| | 0.25 | 0.35 | 0.45 | 0.55 | 0.65 | 0.75 | 0.90 | 1.25 | 2.00 | 3.00 | 3.39 | 4.00 |
| Clean ocean | 1.13 | 1.06 | 1.03 | 1.00 | 0.98 | 0.96 | 0.93 | 0.84 | 0.60 | 0.57 | 0.45 | 0.31 |
| Polluted ocean | 1.41 | 1.22 | 1.09 | 1.00 | 0.94 | 0.89 | 0.82 | 0.70 | 0.48 | 0.47 | 0.36 | 0.24 |
| Arid | 1.12 | 1.07 | 1.03 | 1.00 | 0.98 | 0.96 | 0.95 | 0.92 | 0.82 | 0.66 | 0.59 | 0.50 |
| Clean land | 2.27 | 1.70 | 1.29 | 1.00 | 0.79 | 0.64 | 0.48 | 0.29 | 0.13 | 0.17 | 0.08 | 0.07 |
| Polluted land | 2.29 | 1.71 | 1.30 | 1.00 | 0.79 | 0.64 | 0.48 | 0.29 | 0.13 | 0.17 | 0.09 | 0.07 |
| Highly polluted land | 2.33 | 1.74 | 1.30 | 1.00 | 0.79 | 0.64 | 0.49 | 0.30 | 0.14 | 0.16 | 0.09 | 0.07 |

**Table S7.** Spectral SSA

| Aerosol Type | Wavelength (microns) | | | | | | | | | | | |
|---|---|---|---|---|---|---|---|---|---|---|---|---|
| | 0.25 | 0.35 | 0.45 | 0.55 | 0.65 | 0.75 | 0.90 | 1.25 | 2.00 | 3.00 | 3.39 | 4.00 |
| Clean ocean | 0.96 | 0.99 | 1.00 | 1.00 | 1.00 | 1.00 | 1.00 | 0.99 | 0.99 | 0.45 | 0.88 | 0.97 |
| Polluted ocean | 0.90 | 0.95 | 0.96 | 0.96 | 0.97 | 0.97 | 0.97 | 0.97 | 0.97 | 0.42 | 0.87 | 0.95 |
| Arid | 0.68 | 0.75 | 0.83 | 0.88 | 0.91 | 0.92 | 0.93 | 0.93 | 0.93 | 0.77 | 0.86 | 0.94 |
| Clean land | 0.88 | 0.97 | 0.97 | 0.96 | 0.95 | 0.94 | 0.92 | 0.87 | 0.88 | 0.33 | 0.78 | 0.85 |
| Polluted land | 0.83 | 0.90 | 0.90 | 0.89 | 0.88 | 0.87 | 0.84 | 0.79 | 0.77 | 0.31 | 0.69 | 0.74 |
| Highly polluted land | 0.72 | 0.77 | 0.77 | 0.75 | 0.74 | 0.72 | 0.69 | 0.62 | 0.55 | 0.24 | 0.50 | 0.53 |

**Table S8.** Asymmetry Parameter

| Aerosol Type | Wavelength (microns) | | | | | | | | | | | |
|---|---|---|---|---|---|---|---|---|---|---|---|---|
| | 0.25 | 0.35 | 0.45 | 0.55 | 0.65 | 0.75 | 0.90 | 1.25 | 2.00 | 3.00 | 3.39 | 4.00 |
| Clean ocean | 0.77 | 0.76 | 0.75 | 0.76 | 0.76 | 0.76 | 0.77 | 0.78 | 0.78 | 0.75 | 0.71 | 0.71 |
| Polluted ocean | 0.75 | 0.74 | 0.73 | 0.74 | 0.74 | 0.74 | 0.75 | 0.76 | 0.78 | 0.74 | 0.71 | 0.71 |
| Arid | 0.82 | 0.79 | 0.75 | 0.73 | 0.71 | 0.70 | 0.70 | 0.69 | 0.69 | 0.71 | 0.70 | 0.68 |
| Clean land | 0.73 | 0.71 | 0.69 | 0.68 | 0.67 | 0.66 | 0.64 | 0.62 | 0.71 | 0.76 | 0.76 | 0.78 |
| Polluted land | 0.72 | 0.70 | 0.69 | 0.67 | 0.66 | 0.65 | 0.64 | 0.62 | 0.70 | 0.76 | 0.76 | 0.78 |
| Highly polluted land | 0.70 | 0.68 | 0.66 | 0.65 | 0.64 | 0.63 | 0.62 | 0.61 | 0.69 | 0.75 | 0.75 | 0.78 |

**Comment:**

**3. Page 11 lines 12-15 and page 12, line 2: I don't think this statement about POLDER is accurate. If the authors are using the GRASP POLDER product, the POLDER inversion is multispectral, multidirectional, and multi-polarisation so has good constraints on SSA across the visible. Plus, POLDER did not have any UV bands. It is not clear which POLDER product was used here (this should be stated). If it was not GRASP, then it ideally should have been, as this is to my knowledge the best one that is available at present. I also think more discussion should be given to the big offsets between the data sets over some land regions – the authors may be interested in the Chen et al (2020) paper which evaluates GRASP POLDER retrievals: https://doi.org/10.5194/essd-12-3573-2020**

**Response:** Yes, we have used the GRASP POLDER product. The lines mentioned were written with my limited experience with polarimetric aerosol datasets. We agree the UV band retrievals are only for OMI. Those lines have been changed.

The following lines have been added based on the results from Chen et al. (2020):

**Additions to manuscript:**

**Page 11, lines 15-19:** Large variations in SSA can be observed between CERES-MODIS and POLDER, especially over land where the aerosol loading is less. POLDER SSA retrievals are more accurate for higher aerosol loading. Chen et al. 2020 has shown that POLDER SSA (670 nm) comparison with AERONET significantly improves with correlation coefficient increasing from 0.321 to 0.814, and RMSE decreasing from 0.056 to 0.029 for AOD greater than 1.5.


**Response:** Thank you very much for this suggestion. As mentioned by the reviewer, the comparisons have improved when CERES-MODIS data is compared with interpolated AERONET 550 nm data. The figure and text have been updated accordingly.

[Figure]

**Figure 7.** CERES-MODIS SSA (550 nm) vs. AERONET SSA (550 nm) for various AERONET sites classified based on the type of aerosols (Giles et al., 2012)

**Comment:**

**6. Section 8: I would prefer if this were written in paragraph form, which is more common, but that is more a decision for the Editor and journal style guide.**

**Response:** The conclusion section has been rewritten in paragraph form.

> **Additions to revised manuscript:**
>
> **8. Summary and Conclusions**
>
> Global maps of aerosol absorptions were generated using the newly developed combined CERES-MODIS algorithm based on the concept of critical optical depth. The CERES-MODIS dataset was compared with OMI and POLDER SSA datasets. The retrieved SSA values were also compared with available aircraft measurements over India and surrounding oceanic regions, which showed that most retrieved SSA values are within $\pm 0.03$. We showed that the combined CERES-MODIS algorithm better captures the spatial and seasonal variations in aerosol absorption and the resultant maps provide an improved global SSA database with fewer data gaps. Global mean SSA was estimated to be 0.93 and 0.97 over land and ocean, respectively. The algorithm's sensitivity to various parameters has been studied, which shows maximum sensitivity to changes in surface albedo. The algorithm is shown to be the most effective over regions with large AOD variations and less surface albedo variations. Comparison with SSA from 15 AERONET sites showed an acceptable agreement between AERONET and CERES-MODIS SSA within their uncertainties. These global maps provide valuable input to models for assessing the aerosol-climate impacts on both regional and global scales.

**Comment:**

**7. Page 18 line 14: the authors claim "improved" accuracy over previous techniques to estimate SSA, but I do not see evidence that this is the case? I agree that there are fewer data gaps. The wording should be checked and either backed up with facts or removed (please do not exaggerate findings).**

**Response:** The sentence mentioned has been removed.

[revised manuscript text omitted]
 | 0.91 ± 0.01 (0.92 ± 0.01) [0.93 ± 0.02] | 0.90 ± 0.01 (0.94 ± 0.01) [0.91 ± 0.02] | 0.91 ± 0.02 (0.95 ± 0.01) [0.95 ± 0.02] | 0.91 ± 0.02 (0.94 ± 0.02) [0.93 ± 0.03] |

[Figure]

**Figure S2.** Map showing location of AERONET sites used in this study. The type of aerosols (dust, mixed, urban and biomass) were as defined in Giles et al., 2012

**Table S3**: Name of AERONET site as shown in Fig. S2

| No. | Name | No. | Name | No. | Name |
|-----|------|-----|------|-----|------|
| 1 | GSFC | 6 | Capo_Verde | 11 | SEDE_BOKER |
| 2 | Mexico_City | 7 | Dakar | 12 | Kanpur |
| 3 | Alta_Floresta | 8 | Illorin | 13 | XiangHe |
| 4 | Ispra | 9 | Banizoumbou | 14 | Shirahama |
| 5 | Moldova | 10 | Mongu | 15 | Lake_Argyle |

[Figure]

**Figure S3.** Seasonal mean shortwave-integrated surface albedo from CERES

**Table S4**. Shortwave integrated seasonal mean surface albedo from CERES over regions of interest. Details of these regions are given in Table S1 and Fig. S1

| Region | Surface Albedo | | | |
|---|---|---|---|---|
| | DJF | MAM | JJA | SON |
| Canadian Boreal Forest | 0.36 ± 0.13 | 0.30 ± 0.12 | 0.12 ± 0.03 | 0.16 ± 0.05 |
| Russian Boreal Forest | 0.37 ± 0.10 | 0.27 ± 0.08 | 0.13 ± 0.02 | 0.20 ± 0.05 |
| South African Forest | 0.12 ± 0.01 | 0.13 ± 0.01 | 0.12 ± 0.02 | 0.13 ± 0.01 |
| Amazon Forest | 0.14 ± 0.01 | 0.14 ± 0.01 | 0.13 ± 0.02 | 0.14 ± 0.02 |
| North East Atlantic | 0.06 ± 0.01 | 0.05 ± 0.01 | 0.05 ± 0.01 | 0.05 ± 0.01 |
| South East Atlantic | 0.05 ± 0.01 | 0.05 ± 0.01 | 0.05 ± 0.01 | 0.05 ± 0.01 |
| Eastern Pacific | 0.05 ± 0.01 | 0.05 ± 0.00 | 0.05 ± 0.01 | 0.05 ± 0.00 |
| Sahara | 0.35 ± 0.06 | 0.34 ± 0.06 | 0.34 ± 0.06 | 0.34 ± 0.06 |
| Indo Gangetic Plain | 0.13 ± 0.02 | 0.13 ± 0.02 | 0.14 ± 0.02 | 0.13 ± 0.01 |
| Eastern China | 0.13 ± 0.04 | 0.13 ± 0.03 | 0.13 ± 0.03 | 0.13 ± 0.03 |
| Arabian Sea | 0.06 ± 0.01 | 0.05 ± 0.01 | 0.05 ± 0.02 | 0.05 ± 0.01 |
| Bay of Bengal | 0.05 ± 0.01 | 0.05 ± 0.01 | 0.05 ± 0.01 | 0.05 ± 0.01 |

**Details of aerosol models used**

The aerosol models used are from OPAC (Optical Properties of Aerosols and Clouds), developed by Hess et al., (1998). The existing mixture of aerosol types in OPAC is used – clean ocean, polluted ocean, arid, clean land, polluted land, and highly polluted land.

A LUT is indexed by surface albedo, water vapour, and SSA. The LUT of the aerosol type selected for the pixel is used to compute SSA from $\tau_c$.

[Figure]

**Fig S4.** An inverse look-up is performed to computer SSA from $\tau_c$. Details of the "Identify Aerosol Model" block are shown in Fig S5.

The aerosol type is selected based on geographical location (Ocean/land, surface albedo) and aerosol loading (AOD).

[Figure]

**Fig S5.** (a) Decision tree for selecting the aerosol model. (b) Shows a sample map of the aerosol model used for a particular day 03 Jun 2017, following the color code used in the decision tree

**Table S5.** Components of the mixed aerosol types used

| Aerosol Type | Components | Number density (1/cm^3) | Number mixing ratio | Volume mixing ratio | Mass mixing ratio |
|---|---|---|---|---|---|
| clean ocean | waso | 1.50E+03 | 9.87E-01 | 6.44E-02 | 7.05E-02 |
| | ssam | 2.00E+01 | 1.32E-02 | 9.15E-01 | 9.09E-01 |
| | sscm | 3.20E-03 | 2.11E-06 | 2.03E-02 | 2.01E-02 |
| Polluted ocean | waso | 3.80E+03 | 4.22E-01 | 1.47E-01 | 1.60E-01 |
| | soot | 5.18E+03 | 5.76E-01 | 8.53E-03 | 6.53E-03 |
| | ssam | 2.00E+01 | 2.22E-03 | 8.26E-01 | 8.15E-01 |
| | sscm | 3.20E-03 | 3.56E-07 | 1.83E-02 | 1.80E-02 |
| Arid | waso | 2.00E+03 | 8.70E-01 | 3.19E-02 | 1.77E-02 |
| | minm | 2.70E+02 | 1.17E-01 | 3.26E-02 | 3.31E-02 |
| | miam | 3.05E+01 | 1.33E-02 | 7.35E-01 | 7.46E-01 |
| | micm | 1.42E-01 | 6.17E-05 | 2.00E-01 | 2.03E-01 |
| Clean land | inso | 1.50E-01 | 5.77E-05 | 3.27E-01 | 4.07E-01 |
| | waso | 2.60E+03 | 1.00E+00 | 6.73E-01 | 5.93E-01 |
| Polluted land | inso | 4.00E-01 | 2.61E-05 | 3.15E-01 | 3.96E-01 |
| | waso | 7.00E+03 | 4.58E-01 | 6.53E-01 | 5.83E-01 |
| | soot | 8.30E+03 | 5.43E-01 | 3.29E-02 | 2.07E-02 |
| Highly polluted land | inso | 6.00E-01 | 1.20E-05 | 2.28E-01 | 2.99E-01 |
| | waso | 1.57E+04 | 3.14E-01 | 7.07E-01 | 6.58E-01 |
| | soot | 3.43E+04 | 6.86E-01 | 6.56E-02 | 4.31E-02 |

**

*inso – insoluble*          *minm – mineral (nuclei mode)*
*waso – water soluble*       *miam – mineral (accumulation mode)*
*ssam – sea salt (accumulation mode)*   *micm – mineral (coarse mode)*
*sscm - sea salt (coarse mode)*

*\* More details such as refractive index and size distributions can be referred to in Hess et al 1998*

**Table S6.** Normalized extinction coefficient for each aerosol type

| Aerosol Type | Wavelength (microns) | | | | | | | | | | | |
|---|---|---|---|---|---|---|---|---|---|---|---|---|
| | 0.25 | 0.35 | 0.45 | 0.55 | 0.65 | 0.75 | 0.90 | 1.25 | 2.00 | 3.00 | 3.39 | 4.00 |
| Clean ocean | 1.13 | 1.06 | 1.03 | 1.00 | 0.98 | 0.96 | 0.93 | 0.84 | 0.60 | 0.57 | 0.45 | 0.31 |
| Polluted ocean | 1.41 | 1.22 | 1.09 | 1.00 | 0.94 | 0.89 | 0.82 | 0.70 | 0.48 | 0.47 | 0.36 | 0.24 |
| Arid | 1.12 | 1.07 | 1.03 | 1.00 | 0.98 | 0.96 | 0.95 | 0.92 | 0.82 | 0.66 | 0.59 | 0.50 |
| Clean land | 2.27 | 1.70 | 1.29 | 1.00 | 0.79 | 0.64 | 0.48 | 0.29 | 0.13 | 0.17 | 0.08 | 0.07 |
| Polluted land | 2.29 | 1.71 | 1.30 | 1.00 | 0.79 | 0.64 | 0.48 | 0.29 | 0.13 | 0.17 | 0.09 | 0.07 |
| Highly polluted land | 2.33 | 1.74 | 1.30 | 1.00 | 0.79 | 0.64 | 0.49 | 0.30 | 0.14 | 0.16 | 0.09 | 0.07 |

**Table S7.** Spectral SSA

| Aerosol Type | Wavelength (microns) | | | | | | | | | | | |
|---|---|---|---|---|---|---|---|---|---|---|---|---|
| | 0.25 | 0.35 | 0.45 | 0.55 | 0.65 | 0.75 | 0.90 | 1.25 | 2.00 | 3.00 | 3.39 | 4.00 |
| Clean ocean | 0.96 | 0.99 | 1.00 | 1.00 | 1.00 | 1.00 | 1.00 | 0.99 | 0.99 | 0.45 | 0.88 | 0.97 |
| Polluted ocean | 0.90 | 0.95 | 0.96 | 0.96 | 0.97 | 0.97 | 0.97 | 0.97 | 0.97 | 0.42 | 0.87 | 0.95 |
| Arid | 0.68 | 0.75 | 0.83 | 0.88 | 0.91 | 0.92 | 0.93 | 0.93 | 0.93 | 0.77 | 0.86 | 0.94 |
| Clean land | 0.88 | 0.97 | 0.97 | 0.96 | 0.95 | 0.94 | 0.92 | 0.87 | 0.88 | 0.33 | 0.78 | 0.85 |
| Polluted land | 0.83 | 0.90 | 0.90 | 0.89 | 0.88 | 0.87 | 0.84 | 0.79 | 0.77 | 0.31 | 0.69 | 0.74 |
| Highly polluted land | 0.72 | 0.77 | 0.77 | 0.75 | 0.74 | 0.72 | 0.69 | 0.62 | 0.55 | 0.24 | 0.50 | 0.53 |

**Table S8.** Asymmetry Parameter

| Aerosol Type | Wavelength (microns) | | | | | | | | | | | |
|---|---|---|---|---|---|---|---|---|---|---|---|---|
| | 0.25 | 0.35 | 0.45 | 0.55 | 0.65 | 0.75 | 0.90 | 1.25 | 2.00 | 3.00 | 3.39 | 4.00 |
| Clean ocean | 0.77 | 0.76 | 0.75 | 0.76 | 0.76 | 0.76 | 0.77 | 0.78 | 0.78 | 0.75 | 0.71 | 0.71 |
| Polluted ocean | 0.75 | 0.74 | 0.73 | 0.74 | 0.74 | 0.74 | 0.75 | 0.76 | 0.78 | 0.74 | 0.71 | 0.71 |
| Arid | 0.82 | 0.79 | 0.75 | 0.73 | 0.71 | 0.70 | 0.70 | 0.69 | 0.69 | 0.71 | 0.70 | 0.68 |
| Clean land | 0.73 | 0.71 | 0.69 | 0.68 | 0.67 | 0.66 | 0.64 | 0.62 | 0.71 | 0.76 | 0.76 | 0.78 |
| Polluted land | 0.72 | 0.70 | 0.69 | 0.67 | 0.66 | 0.65 | 0.64 | 0.62 | 0.70 | 0.76 | 0.76 | 0.78 |
| Highly polluted land | 0.70 | 0.68 | 0.66 | 0.65 | 0.64 | 0.63 | 0.62 | 0.61 | 0.69 | 0.75 | 0.75 | 0.78 |

**Response to reviewer's reply**

MS no: ACP-2021-521

We thank the reviewer for the suggestions. We have revised the manuscript following the corrections suggested by the reviewer. This document outlines the reviewer's comments (in **bold-blue**), followed by the author's responses and changes made in the revised manuscript. A marked-up version of the manuscript showing the revisions is appended to this response file.
* * *
**Comment:**
**The authors have addressed my remaining concerns in this version. I recommend publication following these quite minor corrections.**

**Response:** We thank the reviewer for the feedback. We have revised the manuscript following the suggestions.

**Comment:**
**1. Page 9 line 5: I think the first "500" should read "550" (the CERES data are at 550 nm).**

**Response:** Thank you. It's corrected as 550 nm.

**Comment:**
**2. Page 9 line 7: A reference is needed for the POLDER data set used. Ideally a version number and a paper citation. The version number will depend on what the authors are using. The paper should probably be Dubovik 2014 or Chen 2020. Also, I am not sure what "POLDER 1-2" means here, this should be checked and corrected. If the authors are referring to the instrument, from the time period they are using POLDER 3 data and not POLDER 1 or 2.**

**Response:** The "POLDER 1-2" was a typing mistake. It was meant to be "POLDER v1.2". Thank you for mentioning this. It is now corrected. We have also added the relevant citations.

**Additions to the manuscript** (underlined)

Page 9 Lines 7-9: And POLDER v1.2 Level 3 (Dubovik et al., 2011, 2014, 2021) climatological seasonal mean SSA maps are shown in panels b, d, f, and h in Fig 5.

**Response:** Done. All occurences were typset as Ångström.

[revised manuscript text omitted]